# The distribution and abundance of planktonic foraminifera under summer sea-ice in the Arctic Ocean

Flor Vermassen[1,2], Clare Bird[3], Tirza M. Weitkamp[1,2], Kate F. Darling[3,4], Hanna Farnelid[5], Céline Heuzé[6], Allison Y. Hsiang[1,2], Salar Karam[6], Christian Stranne[1,2], Marcus Sundbom[2,7], Helen K. Coxall[1,2]

[1]Department of Geological Sciences, Stockholm University, Stockholm, Sweden
[2]Bolin Centre for Climate Research, Stockholm University, Stockholm, Sweden
[3]Biological and Environmental Sciences, Faculty of Natural Sciences, University of Stirling, Stirling, United Kingdom
[4]School of GeoSciences, University of Edinburgh, Edinburgh, United Kingdom
[5]Centre for Ecology and Evolution in Microbial Model Systems - EEMiS, Linnaeus University, Kalmar, Sweden
[6]Department of Earth Sciences, University of Gothenburg, Gothenburg, Sweden
[7]Department of Environmental Science, Stockholm University, Stockholm, Sweden

*Correspondence to*: Flor Vermassen (flor.vermassen@geo.su.se)

**Abstract.** Planktonic foraminifera are calcifying protists that represent a minor yet important part of the pelagic microzooplankton. They are found in all of Earth's ocean basins and are widely studied in sediment records to reconstruct climatic and environmental changes throughout geological time. The Arctic Ocean is currently being transformed in response to modern climate change, yet the effect on planktonic foraminiferal populations is virtually unknown. Here we provide the first systematic sampling of planktonic foraminifera communities in the 'high' Arctic Ocean – here defined as areas north of 80°N – specifically in the broad region located between northern Greenland (Lincoln Sea with adjoining fjords and the Morris Jesup Rise), the Yermak Plateau, and the North Pole. Stratified depth tows down to 1000 m using a multinet were performed to reveal the species composition and spatial variability of these communities below the summer sea-ice. The average abundance in the top 200 m ranged between 15-65 ind.m$^{-3}$ in the central Arctic Ocean and was <0.3 ind.m$^{-3}$ in the shelf area of the Lincoln Sea. At all stations, except one site at the Yermak Plateau, assemblages consisted solely of the polar specialist *Neogloboquadrina pachyderma*. It predominated in the top 100 m, where it was likely feeding on phytoplankton below the ice. Near the Yermak Plateau, at the outer edge of the pack ice, rare specimens of *Turborotalita quinqueloba* occurred that appeared to be associated with the inflowing Atlantic Water layer. Our results would suggest that the anticipated turnover from polar to subpolar planktonic species in the perennially ice-covered part of the central Arctic Ocean has not yet occurred, in agreement with a recent meta-analysis from the Fram Strait which suggested that increased export of sea-ice is blocking the influx of Atlantic-sourced species. The presented dataset will be a valuable reference for continued monitoring of the abundance and composition of planktonic foraminifera communities as they respond to the ongoing sea-ice decline and the 'Atlantification' of the Arctic Ocean basin. Additionally, the results can be used to assist paleoceanographic interpretations, based on sedimented foraminifera assemblages.

## 1 Introduction

Planktonic foraminifera are unicellular protists that form an important component of the pelagic biome. Their calcareous tests are common in the sediment record and are widely used as tracers of climatic and oceanographic conditions throughout geological time (Schiebel & Hemleben, 2017). To understand the response of foraminiferal communities to climatic change, a thorough understanding of their ecology is required. This is crucial for correctly interpreting temporal and spatial variations of foraminiferal assemblages and their geochemical signatures. One region that is vastly understudied with respect to its

planktonic foraminifera is the high Arctic Ocean (here defined as the ocean areas north of 80°N). With sea-ice retreating rapidly (Meier & Stroeve, 2022) and Atlantic waters increasingly influencing the Arctic domain (a process dubbed 'Atlantification'; Polyakov *et al.*, 2017), pelagic ecosystems are expected to be affected significantly (Brandt *et al.*, 2023). While the footprint of these processes on planktonic foraminifera communities is unknown to date, a recent meta-analysis suggests that the increased export of sea-ice through the Fram Strait is currently playing a key role in blocking the flux of

subpolar planktonic foraminifera towards the high Arctic Ocean (Greco *et al.*, 2022). It is anticipated that once this export ceases (i.e. essentially when the Arctic Ocean becomes seasonally ice-free), subpolar species will be able to rapidly invade the high Arctic Ocean (Greco *et al.*, 2022).

Baseline information and monitoring of planktonic foraminifera is needed to assess the impacts of the unfolding changes, especially in the sea-ice dominated high Arctic Ocean. Yet, knowledge of resident pelagic communities in the remote,

perennially ice-covered regions is minimal. Thus far, most studies have documented living planktonic foraminifera from the Arctic region stemming from areas within, or near, the seasonal sea-ice zone or near sea ice that is exported through the Fram Strait (Carstens *et al.*, 1997; Volkmann, 2000; Darling *et al.*, 2004; Pados & Spielhagen, 2014). These studies show that *Neogloboquadrina pachyderma* and *Turborotalita quinqueloba* are the dominant species in these regions, with occasional traces of *Globigerinita uvula* and *Globigerinita glutinata*. Only two previous studies have reached sites located far into the

perennial ice pack; this includes sampling stations at the Alpha Ridge at ca. 83°N (Bé, 1960) and stations up to 86°N in the Nansen Basin (Carstens and Wefer, 1992). Bé (1960) observed assemblages consisting only of *N. pachyderma*, at abundances ranging 0-2.4 ind.m$^{-3}$ at depths down to 500 m (single net). However, due to the large mesh size of the sampling net (200 µm) used in Bé's study, the results are not directly comparable to more recent work, which emphasizes the need for small net mesh sizes in the Arctic Ocean. Ideally 63 µm mesh is required to sample juvenile *N. pachyderma* and the small subpolar species

such as *T. quinqueloba* and *G. uvula*. This is important because *T. quinqueloba* are particularly small in the Arctic region (Kandiano & Bauch, 2002; Vermassen et al., 2023), and thus could be missed without the correct sampling approach. In general, the extent to which *T. quinqueloba* occurs in the central Arctic Ocean is a topical question. Two earlier plankton tow transects crossing the Nansen Basin found that *T. quinqueloba* decreased from ca. 30% of the standing stock at 81°N to 2-15% at 86°N (Carstens and Wefer; 1992). It was suggested that these individuals may have been advected in the Atlantic Water

flow path, via branches of the West Spitsbergen Current which delivers Atlantic Water to the central Arctic Ocean. Carstens & Wefer (1992) also proposed that the area of reproduction of this species was located further south.

Here, we report the assemblage composition and depth distribution of planktonic foraminifera from eight vertical multinet hauls conducted in ice-covered waters in the high Arctic Ocean in late summer 2021 (*SAS ODEN21*). This is supplemented with ten plankton hauls performed in the Lincoln Sea and adjoining fjords (Petermann Fjord and Sherard Osborn Fjord),

conducted in late summer of 2019 *(RYDER19)*. The aim of our study was threefold. First, we aimed to provide a snapshot of the standing stock of planktonic foraminifera underneath the perennial ice cover (summer sea-ice; Fig. 2) and test whether subpolar species such as *T. quinqueloba* were present.   Second, we investigated the relationship between ambient environmental conditions and the observed planktonic foraminifera distribution patterns in order to gain a better understanding of the factors that control planktonic foraminifera abundance and species composition. Third, we used an automated approach

to extract and analyse morphometric data and explored whether this data could reveal clues regarding the population dynamics and, perhaps, the life history of *N. pachyderma*.

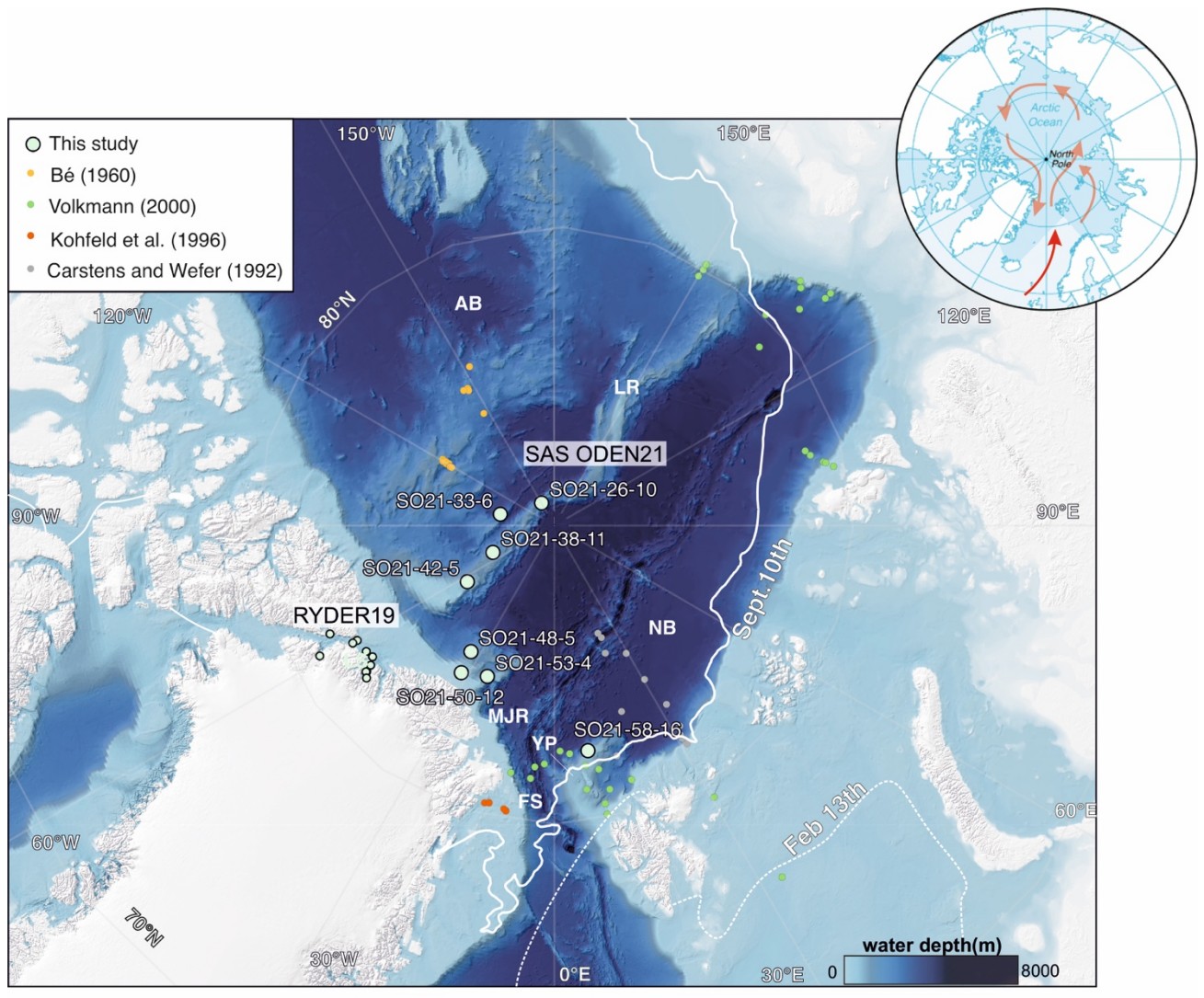

**Figure 1: Bathymetric map of the Arctic Ocean (Jakobsson et al., 2020) with sampling stations obtained during the SAS ODEN 2021 and RYDER 2019 expeditions (this study) and previous foraminifera sampling sites carried out north of 80°N. Full station names and coordinates of sampling sites are listed in Table 1. LR = Lomonosov Ridge, MJR = Morris Jesup Rise, YP= Yermak Plateau, FS = Fram Strait, LS = Lincoln Sea. Inset figure in the upper right shows the track of warm Atlantic waters (red arrows) into the central Arctic Ocean, as they subduct under cooler and fresher waters. Atlantic Water can enter the central Arctic Ocean through the Fram Strait or via the Barents Sea. Sea ice extent on February 13th and September 10th in 2021 are also indicated (data US NIC).**

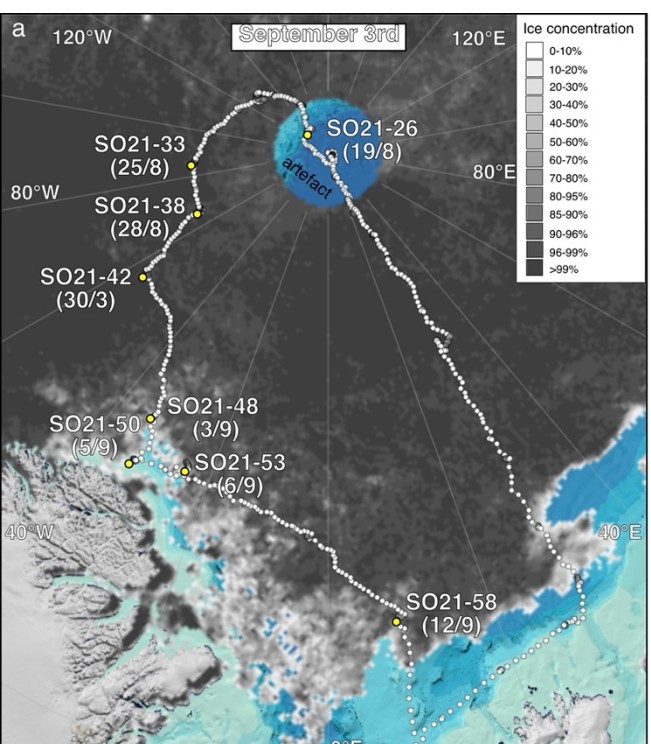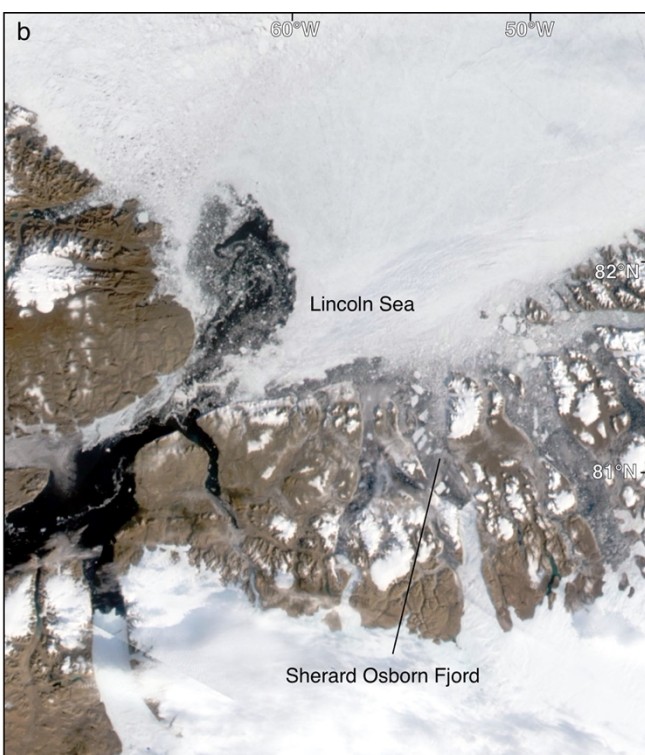

**Figure 2: a) Map displaying sea ice conditions during the expeditions SAS ODEN21 on September 3rd 2021, together with sampling sites. Sea ice data is from University of Bremen, visualised via *https://oden.geo.su.se/map/* . B) Modis satellite image (September 3rd 2019), displaying general ice conditions during RYDER19 expedition. Thick, multi-year sea ice covered most of the Lincoln Sea, whereas open water (with icebergs) was present in the fjords of north Greenland.**

## 2 Methods

### 2.1. Sampling strategy

This study is based on two sampling campaigns conducted in the Arctic Ocean with Icebreaker Oden: the Synoptic Arctic Survey Expedition (hereafter *SAS ODEN21*) and the Ryder 2019 expedition (hereafter *RYDER19*). Multinet sampling was conducted during the *SAS ODEN21* (Snoeijs-Leijonmalm et al., 2022; Fig. 1), whereas single net plankton hauls were conducted during *RYDER19* (Jakobsson *et al.*, 2020; Fig. 1). During *SAS ODEN21*, multinet water column sampling was conducted at eight stations from August 19th to September 11th 2021 at various times during the day under continuous daylight (midnight sun; Table 1). Sample sites include the central Lomonosov Ridge (located 100 km south from the North Pole), the Makarov Basin, the southern Lomonosov Ridge, the area north of Greenland (Morris Jesup Rise), and the Yermak Plateau (Fig. 1). A multinet (Hydro-Bios Multi Plankton Sampler MultiNet®, Type Midi) with surface area of 0.25 m$^2$ was used. The

net mesh was 55 μm, and the mesh of the sampling cup was 50 μm. Net clogging was not an issue in the ice-covered Arctic Ocean where productivity and plankton standing stocks are relatively low compared to other ocean regions.

At each station, the bow of the ship was parked in the ice and plankton nets were deployed from the aft deck, i.e. in areas where the ice had been broken up by the ship. The area where the cable and net enter the water is kept free from incoming ice by water cannons located near the edge of the aft deck – these spray the ocean surface and create a small current away from the ship. Larger pieces of incoming ice were manually diverted using large boathooks to push away incoming ice.

By default, the multinet was lowered to and hauled from 1000 m depth unless the site was shallower, in which case the multinet was hauled from ca. 20 m above the seafloor (Table 1). The default sampling depth intervals were 1000–500 m, 500–200 m, 200-100 m, 100–50 m, and 50–0 m. The upwards towing speed was 0.5 m/s, and towing was paused for approximately one minute at the start of each sampling interval. After deployment, the samples were transferred from the sampling cups to containers. Samples were then pipetted onto a glass petri dish and visualised under the microscope (ZEISS Stemi 508). Planktonic foraminifera individuals were picked onto microfossil slides using a combination of pipettes and brushes. Samples that could not be picked shipboard due to time constraints were preserved in an EtOH solution and picked post-cruise at Stockholm University. All planktonic foraminifera individuals were identified and counted. A systematic counting of cytoplasm content was performed at one station (SO21-26-10), but was not possible at other sites due to time constraints. Throughout the study, we assumed that cytoplasm-bearing individuals were alive, i.e. 'living', and that the empty tests represented dead individuals that were sinking. However, recently deceased individuals may still have cytoplasm, and planktonic foraminifera can exhibit dormancy (Murray & Bowser, 2000; Ross & Hallock, 2016; Westgård *et al.*, 2023) . The abundance of planktonic foraminifera (number of individuals per cubic meter filtered water) was calculated via the formula:

$$\frac{n}{a \cdot d}$$

where n = number of individuals counted for a given depth interval, a = the surface area of the net, and d = the length of the sampled depth interval. The reported abundances include all tests, regardless of cytoplasm content.

A smaller scale sampling program involving more opportunistic sampling was conducted during *RYDER19* held during August 9th - September 3rd 2019 (Table 1; Jakobsson *et al.*, 2020). A simple plankton net (60 cm diameter net opening, 83 μm mesh), with a 13 kg weight attached, was lowered to a depth of 300 m and hauled vertically at a rate of ca. 0.2 m/s, sampling the entire depth interval at ten stations. On deck, the net content was washed out of the sampling cup with filtered seawater into a storage container and foraminifea were picked from the concentrate. For comparison with our data, abundances of *N. pachyderma* in the Nansen basin reported by Carstens and Wefer (1992) were calculated by multiplying their reported % of *N. pachyderma* with their reported numbers of total planktonic foraminifera abundance. The abundance of *N. pachyderma* in the top 200 m of other studies were calculated from the original data available online.

## 2.2. Test size and morphometric analysis

Picked individuals (N=15 381) were imaged with a Leica M205 C microscope – one photo was taken for each square on the microfossil slide. Individuals were segmented from these images using the 'segment' module of AutoMorph (Hsiang *et al.*, 2017; development version available at https://www.github.com/ahsiang/AutoMorph), which automatically separates and extracts individual imaged objects using traditional image processing methods. As the lighting conditions were set manually and could change between images, all images were initially processed under 'Sample' mode in AutoMorph to determine the optimal parameters (i.e., threshold value and min./max. size of legitimate objects) for segmenting objects in each image. Threshold values tested ranged from 0.10 to 0.79 and size ranged from 50 to 500 μm depending on individual image conditions. Optimal parameter values were chosen to maximize the number of individuals correctly identified. A pixel size of 2.571 μm was used for both the x- and y-axes based on output from the Leica microscope. The images were then processed under 'Final' mode with the optimal parameter values using batch-processing mode to obtain individual cropped images of each specimen. Incorrectly identified non-foraminifer material (e.g., organic fluff and junk images of the background texture of the slides) were removed manually in post-processing. All specimen images were compiled and then processed using the 'run2dmorph' module of AutoMorph. This module detects the outlines of individuals and then automatically extracts morphometric measurements such as major/minor axis length, perimeter length, area, etc. Default values were used for all input parameters to 'run2dmorph'.

## 2.3. CTD, chlorophyll *a* and nutrients

Conductivity-Temperature-Depth (CTD) measurements for *SAS ODEN21* were obtained using two standard Seabird SBE911 plus systems, one "shallow" (maximum 1000 m depth) and one "deep" (full-depth), each with dual sensors to measure temperature and salinity, and single sensors to measure pressure and dissolved oxygen concentration. For more information about hydrographic operations, we refer to the cruise report provided by Snoeijs-Lejonmalm et al. (2022). All sensors have been pre- and post-cruise calibrated by the Swedish Polar Research Secretariat. Salinity and Oxygen were further calibrated against sample data collected from the bottles. Salinity samples from the deep stations were analysed post-cruise using a Guildline Autosal salinometer and IAPSO standard seawater at the GEOMAR, Germany. Dissolved oxygen was determined onboard using an automatic Winkler titration setup with UV detection (Scripps Institute of Oceanography Oxygen Titration System version 2.35m).The fully calibrated data sets are freely available at the PANGAEA database (Heuzé et al., 2022a; sensor data) and (Heuzé et al., 2022b; bottle data).

CTD measurements during the Ryder 2019 expedition were made with a Seabird SBE911 plus with dual temperature (SBE 3) and conductivity (SBE 04C) sensors. The CTD was equipped with a dissolved oxygen sensor (SBE 43), turbidity sensor (WET labs ECO), fluorescence sensor (WET Labs ECO-AFL/FL) and altimeter (BENTHOS ALTIMETER PSA-916D). The CTD was mounted on a 24 Niskin bottle (12 liters) rosette. All sensors were pre- and post-cruise calibrated by the Swedish Polar Research Secretariat. The data set is available at the Bolin Centre Database (Stranne et al., 2020).

On the *SAS ODEN21* expedition, seawater was collected using a SBE rosette system equipped with 22 Niskin bottles (12 L), that was deployed from the bow of the ship. The bottles were closed at predefined depths (10, 20, 30, 40, 50, 75, 100, 125, 150, 200, 250, 300, 400, 500, 700, 1000 m, following the international SAS protocol) during the return of the CTD from the bottom. Directly after the CTD was back on deck, water samples (100 ml) were taken from each Niskin bottle using clean sampling methods. As soon as possible after sampling, typically on the same day, concentrations of the macro-nutrients phosphate, nitrate+nitrite, ammonium, and silicate ($PO_4$, $NO_3+NO_2$, $NH_4$, and $SiO_4$) were determined colorimetrically on board using a four- channel continuous flow analyser with photometric detection (QuAAtro39, SEAL Analytical) on unfiltered seawater. The instrument was set up to use QuAAtro Methods No. Q-064-05 Rev. 8, Q-119-11 Rev. 2, Q-069-05 Rev. 8 and Q-066-05 Rev. 5 for $PO_4$, $NO_3+NO_2$, $NH_4$, and $SiO_4$, respectively. These methods largely correspond to standard methods SS-EN ISO 15681-2:2018, SS-EN ISO 13395:1996, SS-EN ISO 11732:2005 and SS-EN ISO 16264:2004. Each analysis run included standards freshly prepared from stock solutions, certified reference material (VKI QC SW4.1B and VKI QC SW4.2B) and solutions for automatic baseline/drift correction.

Chlorophyll *a* was sampled at the defined depths down to 500 m. Samples were kept in 4,7 L brown bottles at low (~4˚C) temperatures until processing. Size fractionated samples were attained using 2.0 µm polycarbonate filters (diameter 25 mm) as well as 0.3 µm glass fibre filters (Advantec®, diameter 25 mm). Filters were placed inside Swinnex capsules and serially connected at the end of a peristaltic pump system. Seawater was divided into two 2 L bottles to collect replicate samples and was pumped through the system at a low pump rate (30 rpm) to ensure cell integrity on the filters. Seawater was filtered either until the filter system clogged or 2 L of seawater passed through. Filters were immediately placed in test tubes with 2.5 mL 95% EtOH and stored in the dark at room temperature for >16 hours before measurement on a Trilogy Fluorometer (Turner, USA). The instrument was calibrated using a standard from Anacystis nidulans (Sigma-Aldrich).

## 2.4. Statistical analysis

Statistical analyses were performed using Python (version 3.8.1). The *statsmodels (*version 0.11.1) and *scikit-learn* (version 0.23.1) libraries were employed to explore the data using Generalized Linear Models (GLMs). The input variables included temperature, salinity, chlorophyll concentration, and distance to the sea-ice edge, with values averaged according to the respective depth intervals (N=24). The response variable was the foraminifera abundance. Data from the top three depth intervals (0–50 m, 50–100 m, 100–200 m) were used, as nearly all living (cytoplasm-bearing) individuals were found at those depths. Given that the data were continuous and positively skewed, a gamma distribution with a log link were applied in the GLM. Predictor variables were standardised prior to the analysis. Multicollinearity was evaluated using the Variance Inflation Factor (VIF). All variables exhibited VIF values below 5, indicating that multicollinearity was not a concern.

# 3 Results

## 3.1. Environmental conditions

### 3.1.1. Oceanography

As expected, three distinct water masses were present in the upper 1000 m analysed here (e.g. Rabe et al. 2021; Rudels 2000). In the top 10-150 m, a cold (ca. -1.5°C) and low salinity (29-34 g/kg) water mass was present (Figs. 3 and 4). This layer is commonly referred to as the Polar Surface Water (PSW), defined as $\sigma_\theta < 27.7$ kg m$^{-3}$ (Rudels *et al.*, 2008). Below this, a 500-800 m thick water mass occurred – the Atlantic Water – which is characterised by higher temperatures (>0°C) and relatively high salinity (>34.9 g/kg). This water mass is derived from North Atlantic currents, which subside beneath the cold polar waters as they enter the ice-covered central Arctic Ocean. At station SO21-26-10, in the North Pole area, the Atlantic water layer was markedly warmer ($T_{max}$=1.48°C) than at the other sites on the Lomonosov Ridge and Morris Jesup Rise, implying a more proximal branch of inflowing Atlantic Waters (Figs. 3 and 4). At station SO21-58-16, the PSW was much thinner and Atlantic-derived waters ($T_{max} = 1.64$°C) were present between 100-750 m depth (Fig. 4). Below the Atlantic Water lied the deep water, characterized by temperatures lower than 0°C but with a salinity that remained high.

Similar hydrographic conditions were generally found in the area north of Greenland. In the Sherard Osborn Fjord, however, the shallowest 10 m of the PSW was 'dammed' by the heavy sea-ice conditions outside the fjord, in the Lincoln Sea, which led to highly elevated temperatures (reaching 4 °C) low salinities (< 15 g kg$^{-1}$), and low associated Chl-a concentrations within the fjord (Stranne et al., 2021). The Atlantic Water in the north Greenland fjords is to some extent influenced by subglacial runoff and melting from marine-terminating glaciers forming glacially-modified water. This was particularly evident inside the Sherard Osborn Fjord where local bathymetry influences the deep water circulation inside the fjord (Jakobsson *et al.*, 2020b; Nilsson *et al.*, 2023).

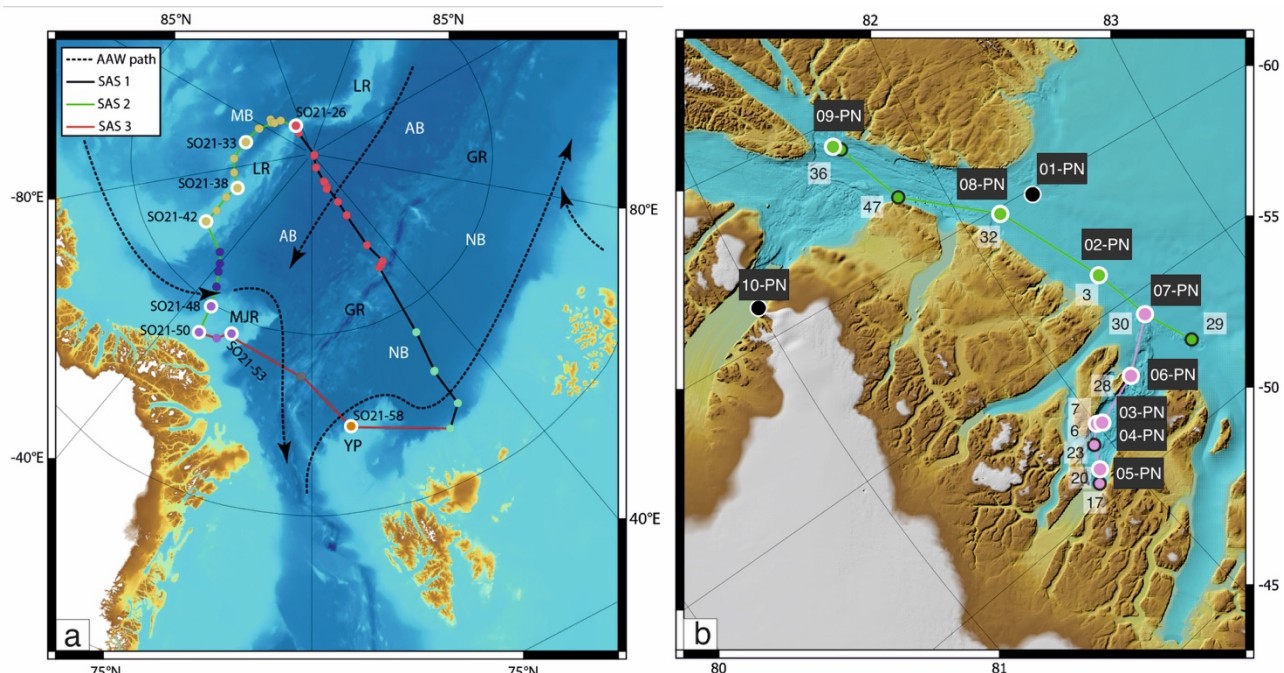

Figure 3: A) Map showing locations of CTD stations during SAS ODEN21 and transects depicted in Figure 4. Sites with white circles depict combined CTD and foraminifera sampling. B) Map showing locations of foraminifer sampling stations (white text, black label: "xx-PN") and CTD stations (black text, white label) during RYDER19 and transects depicted in Figure 4. Sites with white circles depict combined CTD and foraminifera sampling sites. Arrows depicted circulation of Atlantic water.

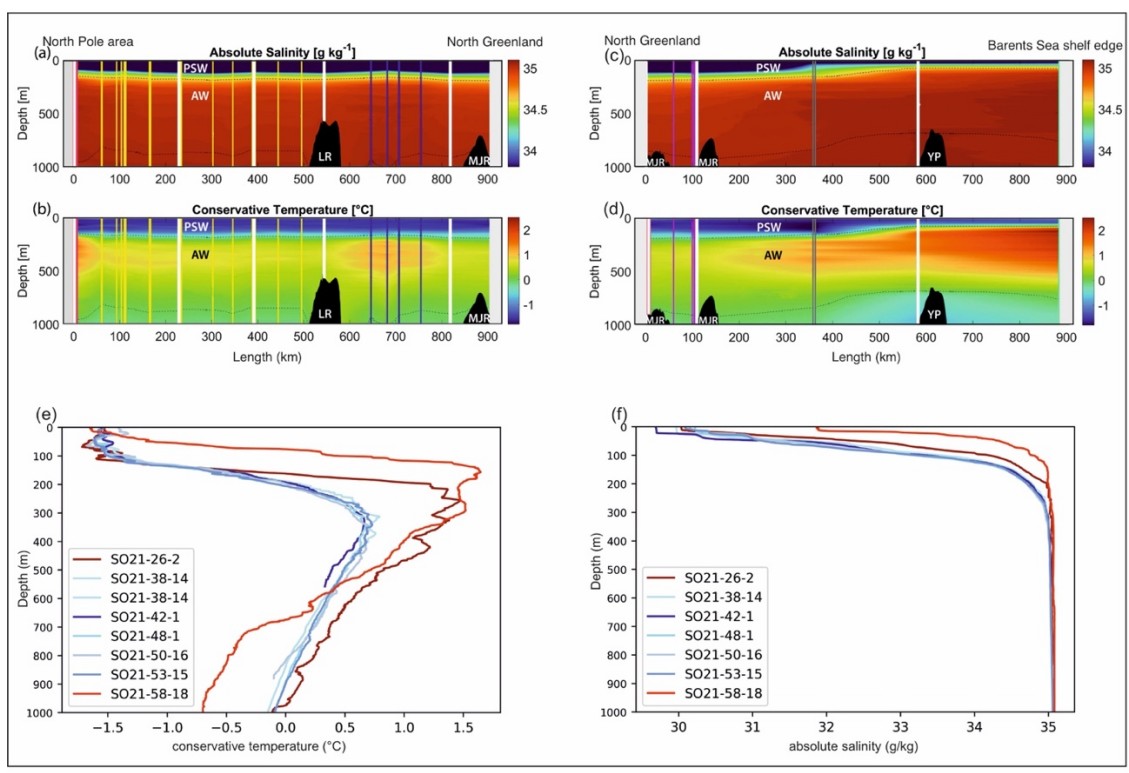

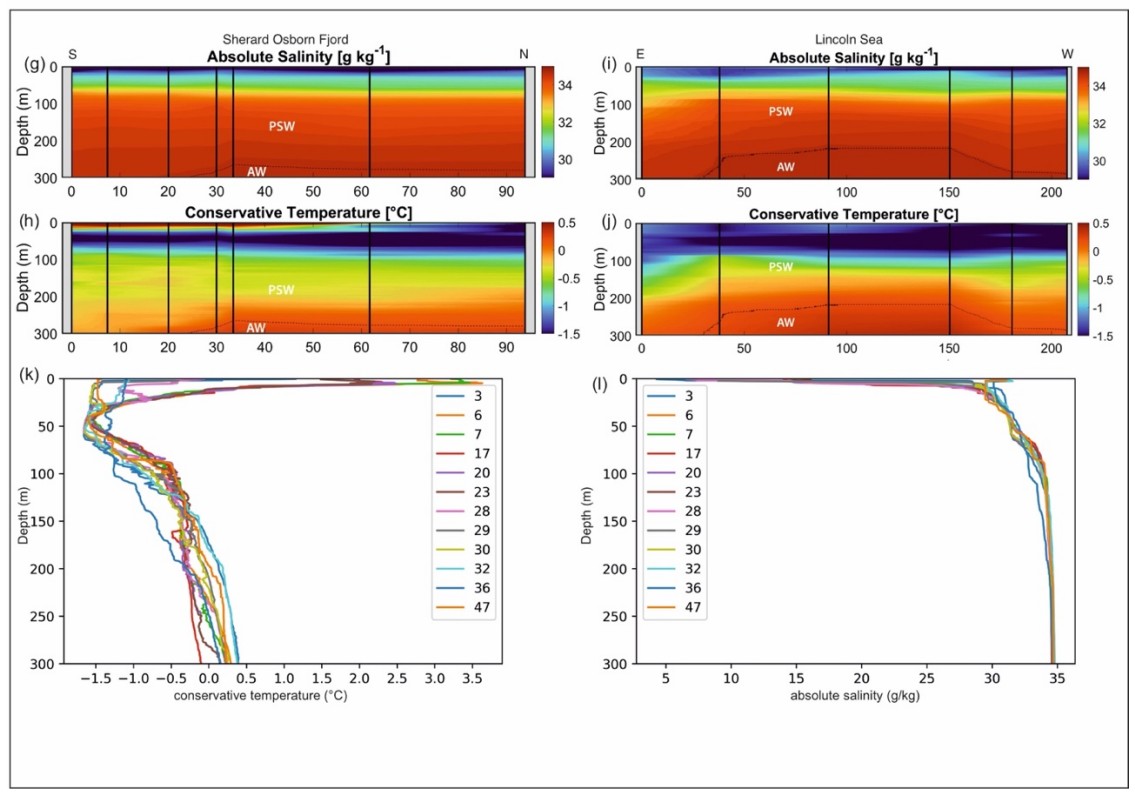

**Figure 4: Oceanographic data (temperature and salinity) obtained during SAS ODEN 21 (a to f) and *RYDER19* (g to l). Transects of temperature (b,d,h,j) and salinity (a,c,g,i) are presented. Temperature profiles (e, k) and salinity profiles (k,l) combined for each expedition are also presented. Salinity and temperature profiles are shown separately for the Lincoln Sea region and Sherard Osborn Fjord in Figures A2 and A3.**

### 3.1.2. Sea ice

All of the stations sampled during *SAS ODEN21* were characterised by intense sea–ice conditions (ice coverage >95 %) (Fig. 2). Sea-ice thickness estimates derived from ice cores obtained near the sampling stations samples ranged from 1.1 to 2.6 m (average of 1.8 m; Snoeijs-Leijonmalm, 2022). However, these observations exhibit a bias towards greater ice thickness as sites with thicker ice were selected for safety precautions and ship stability. Therefore, 'real' regional average ice thickness reported here should be considered significantly lower.

Another factor that has a possible effect on planktonic foraminifera abundance is the distance to the sea-ice edge. For *SAS ODEN21*, Stations SO21-26-10, SO21-33-6, SO21-38-11, SO21-42-5 were located well within in the Arctic ice pack, > 300 km from the nearest sea ice edge, while stations SO21-50-12, SO21-53-4, and SO21-58-16 were located rather close to the ice edge (<50 km). While station SO21-48-5 was located relatively close to a narrow lead of open water, it was located about 80 km from the broader sea-ice edge/marginal ice zone (Fig. 2).

In the Lincoln Sea the ice cover generally consisted of very thick multi-year sea-ice, but areas bordering North Greenland and off Ellesmere Island were temporarily ice-free (Fig. 2). Sherard Osborn was free of sea-ice during the time of sampling, but with icebergs present.

### 3.1.3. Patterns of chlorophyll, nutrient and oxygen concentrations.

Concentrations of chlorophyll *a* were typically moderately high near the surface, increased with depth, and reached a maximum
between 20-40 m water depth (Fig. 5). Chlorophyll *a* maximum values ranged between 0.11-1.10 µg/L. Below this, concentrations decreased, until they became negligible at about 50-70 m depth. At station SO21-58-16 no distinct sub-surface chlorophyll *a* maximum was observed, instead, values were highest near the surface (top 10 m).

Concentrations of nitrate + nitrite ($NO_{23}$), phosphate ($PO_4$) and silicate ($SiO_4$), displayed comparable depth profiles overall, although there were some differences among stations in the top 200 m. From the surface and down to chlorophyll maximum,
nutrients were strongly depleted, especially of $NO_{23}$ (Fig. 4). Between 50-150 m, a pronounced peak occurred at stations SO21-33 to SO21-53, typically with a maximum at around 75 m depth. This peak was much weaker at station SO21-26 where $PO_4$ and $SiO_4$ instead showed a minimum at around 100 m. At the Yermak plateau (station SO21-58-16), there was no subsurface maximum. Rather, there was a steep increase in nutrients concentrations down to 100 m. At greater depths, below 200 m, nutrient concentrations at all stations converged to very similar values.

Stations SO21-33 to SO21-53 shared similar trends in oxygen concentration throughout the water column (Fig. 5). In the top 30 to 50 m of these stations, oxygen values stayed broadly constant around 8.9 ml/l. Down to 100 m, values decreased rapidly

and remained constant at around 7.8 ml/l. At stations 26, values declined from 8.7 ml/l near the surface to 6.9 ml/l at 100 m, between 100-120m values peaked to 7.6 ml/l again and then remained constant at 6.9 ml/l below 120 m. At station SO21-58, oxygen values declined from 8.6 near the surface to 6.9 ml/l at 100 m, below which they remained constant.

The nutrient and oxygen profiles (Fig. 4), as well as the salinity profiles (Fig. 3), indicate the presence of the Transpolar Drift Stream at stations SO21-33 to SO21-53, whereas surface waters near the North Pole (station SO21-26) and particularly on the Yermak Plateau (station SO21-58) were less influenced by this major ocean current in 2021.

## 3.2. Planktonic foraminifera

### 3.2.1. Spatial variability

The variability of average planktonic foraminifera abundance in the top 200 m was relatively high and varied between 18 and 65 individuals/m$^3$ at sites located in the central Arctic Ocean (*SAS ODEN21*; Fig. 6). No obvious spatial trends in abundance were observed in the central Arctic Ocean data set. The abundance of individuals in the water column appeared to be relatively low in the North Pole area (18 ind./m$^3$; station SO21-16-10) and close to the north Greenland coast (19 ind.m$^{-3}$; SO21-50-12, ca. 60 km north of Cape Morris Jesup). The highest abundances occurred at the southern end of the Lomonosov Ridge
(Greenland side) and the northern tip of the Morris Jesup Rise (58 and 65 ind./m$^3$ at SO21-42-5 and SO21-48-5, respectively; Fig. 6). In the Lincoln Sea, Ryder Fjord and Petermann Fjord, abundances were extremely low, with maximum abundances of ca. 0.3 ind.m$^{-3}$ (Fig. 6). Although generally low, it is noteworthy that abundances in the shelf area of the Lincoln Sea were higher than within the fjords. Near the front of Ryder glacier (station RYDER19-05PN), zero individuals were found.

### 3.2.2. Depth variability

Overall, planktonic foraminifera abundances were by far the highest in the top 50 m of the water column (Fig. 5). At five out of eight stations in the central Arctic Ocean, the abundance of individuals in the top 50 m was more than double the abundance of individuals in the 50-100 m interval (Fig 5). At station SO21-33-6, the abundance of individuals was higher in the 50-100m depth interval than in the top 50 m (48 vs 30 ind.m$^{-3}$) and at SO21-38-11, the difference in abundance between the top 50 m and the 50-100m depth interval was small (42 vs 36 ind.m$^{-3}$). At all stations, except SO21-58-10, the percentage of individuals
present in the top 100 m was >65%. At all stations, depths >200 m recorded a low number of individuals (<5 ind./m$^3$), comprising only a minor percentage of the total foraminiferal standing stock at each site. At station SO21-58-16, the depth distribution of foraminifera was different, with the maximum abundance of foraminifera present at 100–200 m, instead of in the top 100 m, with only 28% of the individuals present above 100 m and 72% living below 100 m. Below 500 m, the abundance ranged between 0.03-3.75 ind.m$^{-3}$ (Fig. A1, not shown in Fig. 5 to facilitate better visual comparison with water
column parameters).

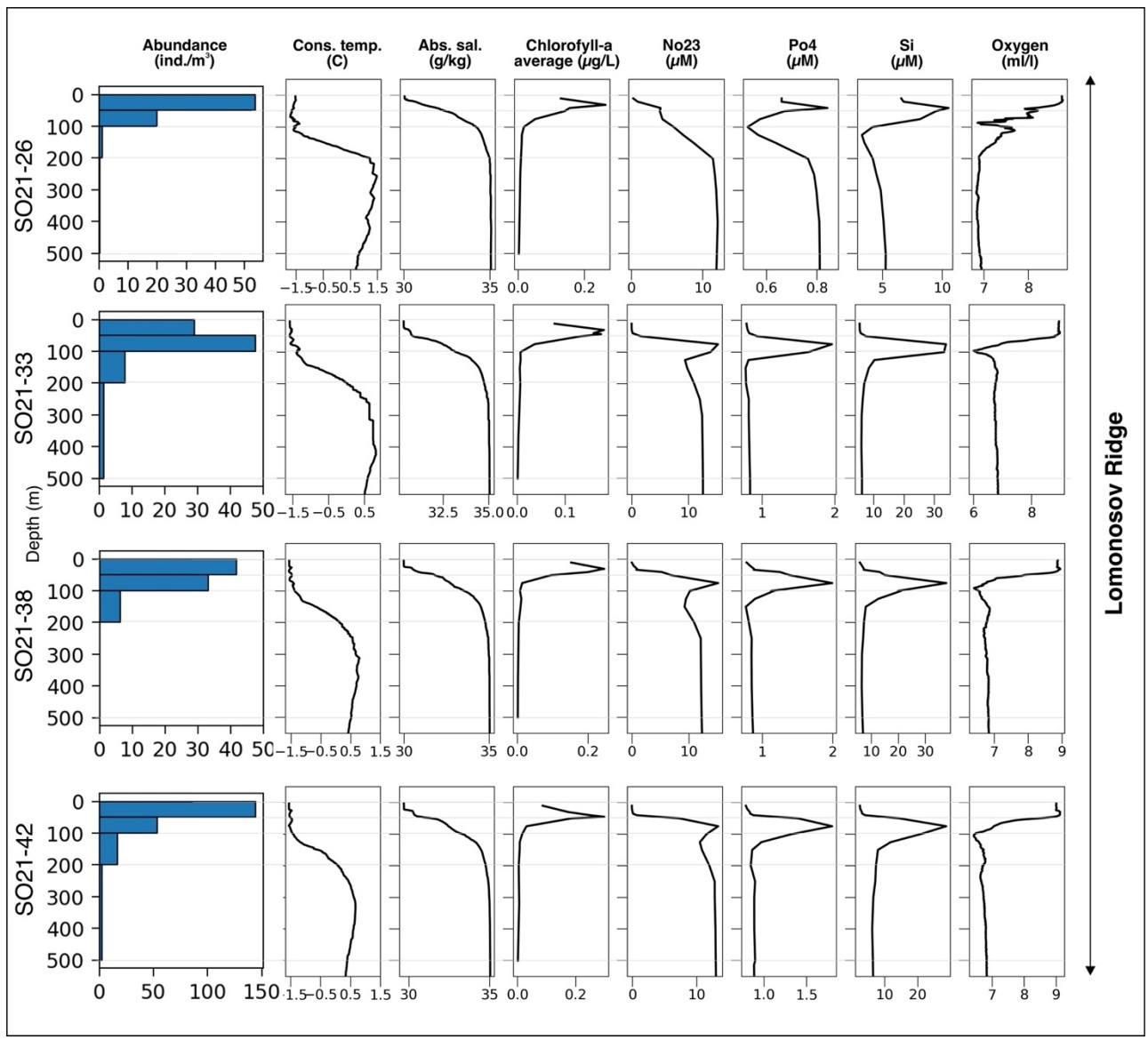

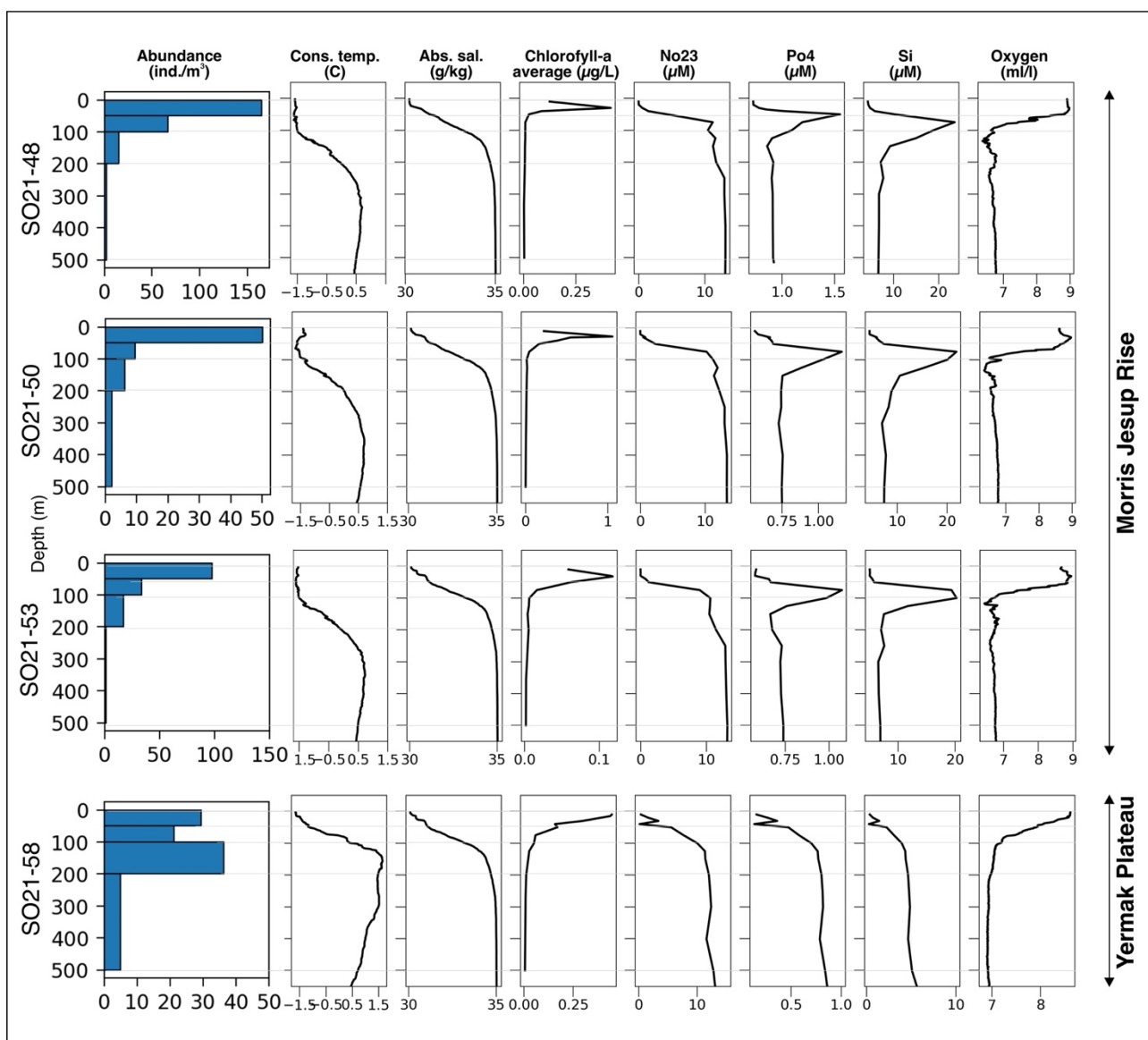

**Figure 5. Overview of planktonic foraminifera abundance in relation to environmental parameters at sampling station (SAS ODEN21). Thin horizontal grey lines indicate the limits of the sampled depth intervals. Blue bars represent abundance of *N. pachyderma*. At site SO21-58, *T. quinqueloba* was found at very low abundances below 100m, and the abundances were lower than the line width of the bars. Note scale change of foraminifer abundance at different sites.**

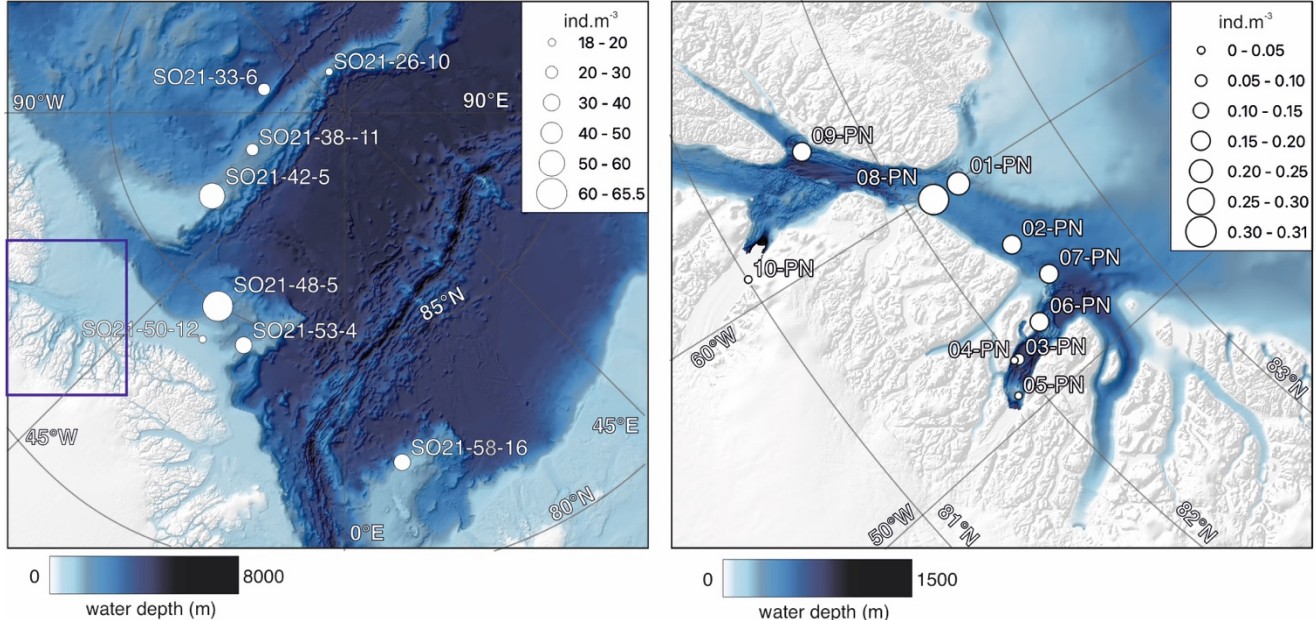

**Figure 6: Left) Average abundance of planktonic foraminifera (ind.m⁻³) in the top 200m for samplings stations obtained during**
**SAS21. Thin box indicates general sampling area of RYDER19. Right) Average abundance of foraminifers (ind.m⁻³) in the top 300m**
**for sites sampled during RYDER 2019, except for stations 04 and 05, where sampling depths reached 408 and 250 m, respectively.**
**Note that no individuals were found at station 05-PN. Note the difference in scale of the abundances between the two panels.**

### 3.2.3 Species composition and size distribution

At all stations, except SO21-58-16, the planktonic foraminifera assemblage was monospecific, consisting of *N. pachyderma*
(Plate 1). At station SO21-58-16, a minor proportion of *T. quinqueloba* was encountered below 50 m, comprising 0.3%, 3.3%
and 3.90% of the assemblage at the 50-100m, 200-500, and 500-1000m depth intervals, respectively (Plate 2). Specimens from
*N. pachyderma* species were pristine and showed no signs of dissolution (Plate 1).

All *N. pachyderma* morphotypes ('Nps 1-5'; Eynaud *et al.*, 2009) were observed (Plate 1). However a deeper analysis of
morphotypes (e.g. morphotype distribution per depth interval), will be the topic of a follow-up study. In the relatively shallow
water depths (the top 100 m) the majority of *N. pachyderma* were small (range of the mean maximum diameter in the top 50
m = 124-141 µm; average of all means in the top 50 m = 134 µm; Fig. 11) and lightly calcified, giving them a translucent
appearance under a light microscope, with an appearance similar to 'Nps-5'. At the deeper levels (>200 m), specimens appeared
to mostly belong to the more heavily calcified morphotypes 1 to 4 (see Introduction and Plate 1). The range of mean maximum
diameter was164-261 µm in the water depths below 500 m, with an average of all means of 202 µm (Fig. 11).

At station SO21-26-10 cytoplasm-bearing shells were observed at all depths but were predominant in the top 100 m (> 75%,
Fig. 8b), whereas tests below 200 m were mostly 'empty', i.e., they were white tests that were settling to the seafloor following
reproduction (<55%, Fig. 8b). Interestingly, the cytoplasm-bearing tests consisted of two types: red and yellow-green (Fig. 8).

Some individuals also had both red- and green-coloured chambers (Fig. 8). The cytoplasm colours transformed rapidly and over time (ca. 12 hours) the relatively bright colours had faded sufficiently to hinder their discrimination. At this site, tests could be rapidly picked and the colour of their cytoplasm was noted immediately (Fig. 9B). This revealed that the red-coloured individuals were more abundant in the top 50 m (64% of cytoplasm-bearing tests were red-coloured at 0-50 m depth), while the green/yellow type were more abundant at 50-100 m (40% were red-coloured). At 100-200 m and 200-500 m, the percentage of tests with red cytoplasm was 60 and 67% respectively, but it should be noted that these numbers are rather uncertain, due to the lower number of tests at these deeper depths (i.e. below 100 m). Due to time constraints, a quantitative assessment of cytoplasm content could not be made at other sites.

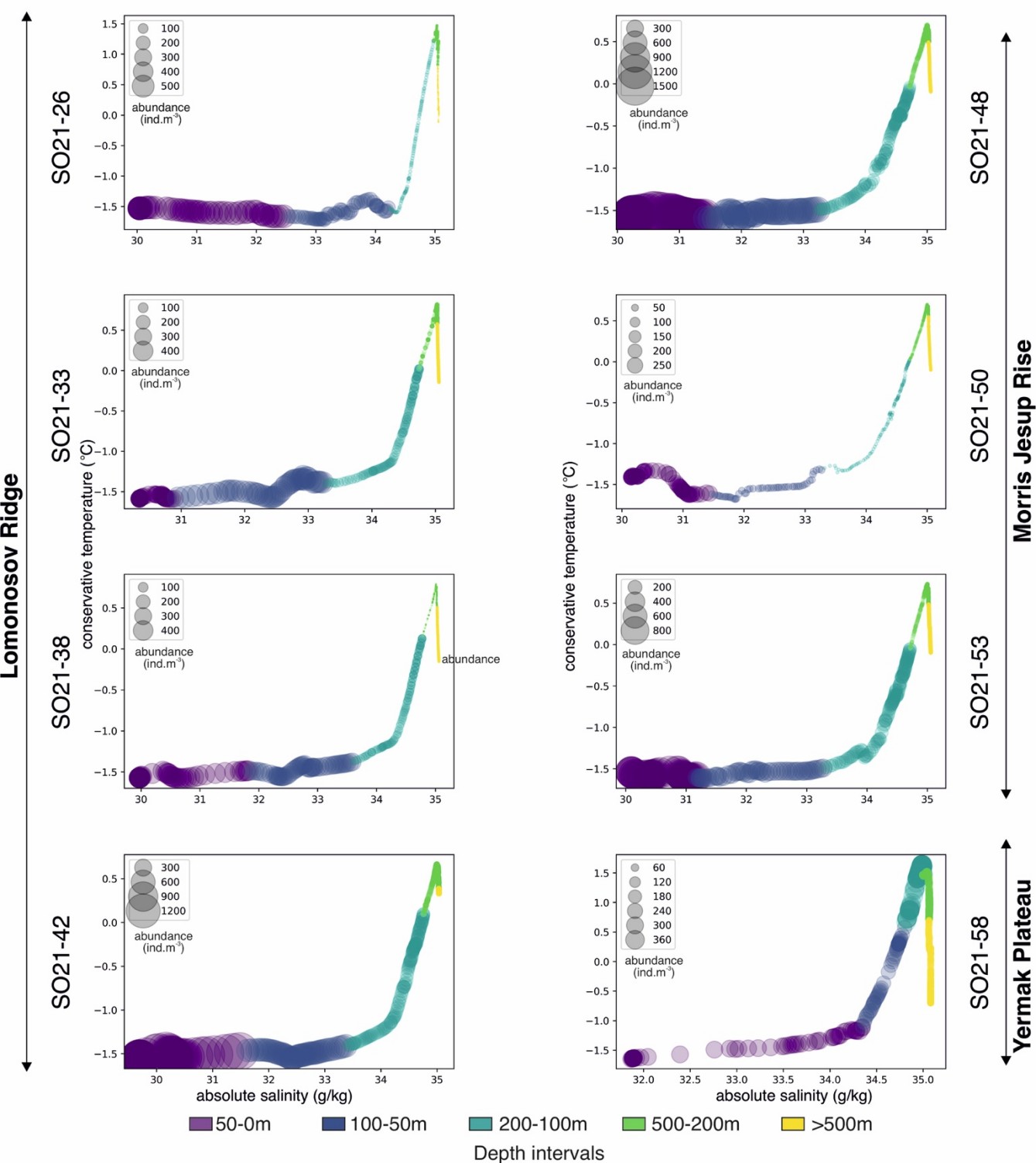

**Fig. 7. Planktonic foraminifera abundance (individuals per m³) plotted in the temperature-salinity space (*SAS ODEN21*). For each T-S data point, the average foraminifer abundance of the corresponding depth interval is plotted. Sizes of the circles represent abundance, colours indicate depth intervals.**

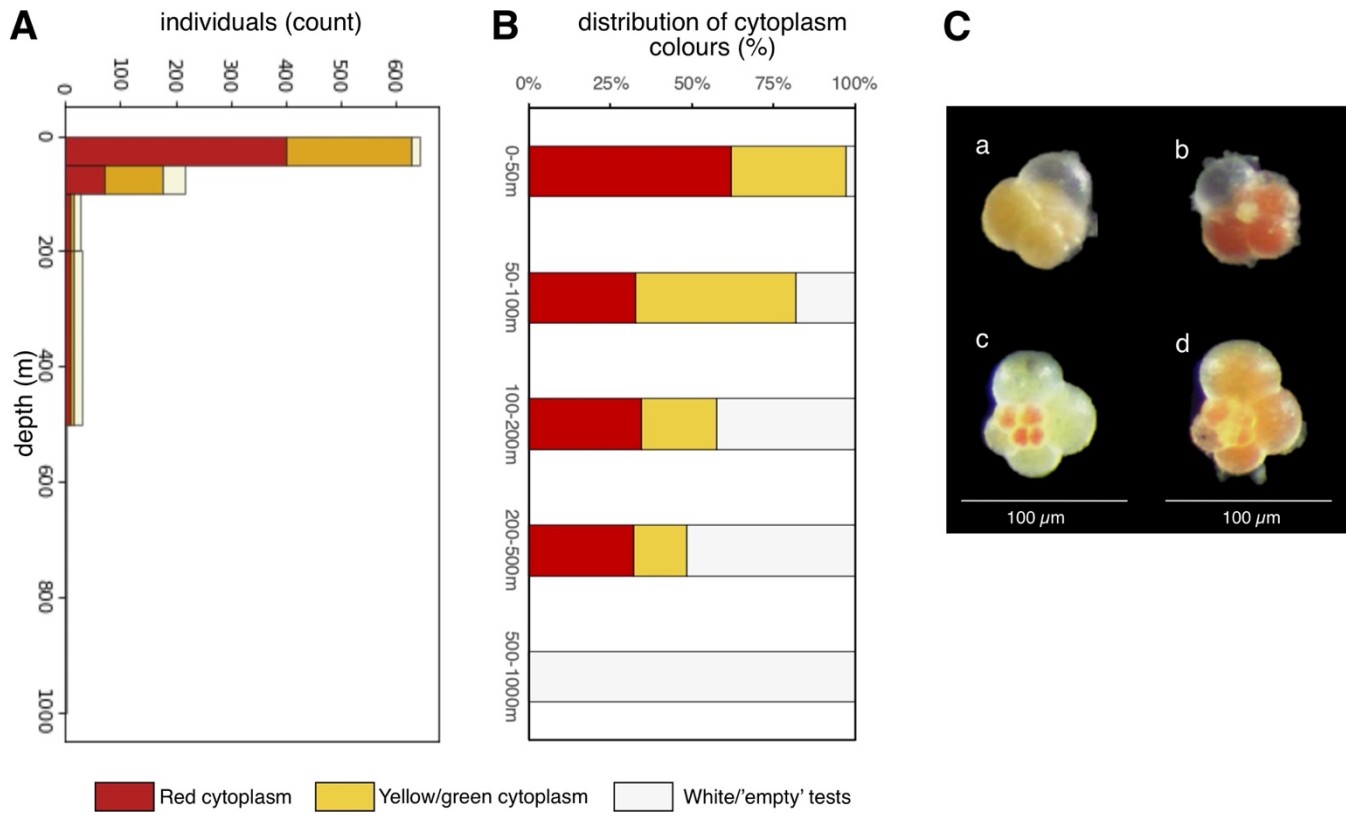

**Fig. 8. A) Counts of individuals at station SO21-26-10, according to cytoplasm colour. B) Plot of relative distribution of tests with different cytoplasm content. C) Example of (a) yellow/green, (b) red, and (c and d) mixed yellow/green-red coloured cytoplasm in living *N. pachyderma*.**

## 4 Discussion

**4.1. Current and future composition of planktonic foraminifera in the central Arctic Ocean**

Our results show that *N. pachyderma* is the only species that currently lives and thrives beneath summer sea-ice in the region between the North Pole and North Greenland. A few occurrences of *T. quinqueloba* were found at the northern tip of the Yermak Plateau (SO21-58-16), proximal to the marginal ice zone at depths below 100 m (<4%*)*. Our results are as expected, since *N. pachyderma* has long been considered the only true polar species that is capable of living in the ice-covered Arctic

Ocean (Bé, 1960; Carstens & Wefer, 1992). *Turborotalita quinqueloba*, on the other hand, is known as a sub-polar species

that thrives in areas near the sea-ice edge, but is not known to reproduce under permanent ice, although it can survive for a limited time in ice-covered conditions (Carstens & Wefer, 1992; Volkmann, 2000; Zamelczyk *et al.*, 2021). In the study of Carstens and Wefer (1992), a significant number of *T. quinqueloba* were found in the Nansen basin (up to 55% south of 83°N and up to 15% north of 83°N), but these individuals were considered to have been advected along with Atlantic currents, with their reproduction area being further south. More recently, a population of *T. quinqueloba* was observed underneath the growing winter sea-ice of the seasonally ice-free Barents Sea in December (comprising 16–67% of the standing stock, the rest being *N. pachyderma*) albeit at very low absolute abundance (<1.5 ind.m$^{-3}$; (Zamelczyk *et al.*, 2021).

The Barents Sea is a hotspot for 'Atlantification' (Polyakov *et al.*, 2017)  and it was suggested that the *T. quinqueloba* population under the winter ice in the Barents Sea was probably not reproducing *in situ*, but rather had stayed in place after reproduction in open waters and the subsequent onset of winter freezing.  From these two studies, it could be anticipated that the rare *T. quinqueloba* occurrences we observed near the Yermak Plateau – where Atlantic waters are present at relatively shallow water depths (T > 0°C at 70 m) – can be explained as individuals that survived under the sea-ice but were not actively reproducing.

The absence of *T. quinqueloba* at sites located near the Lomonosov Ridge and north Greenland – located at higher latitude and/or further along the path of Atlantic currents compared to Carstens and Wefer (1992) – demonstrates that *T. quinqueloba* (or other sub-polar species) are not yet present in the central Arctic Ocean and do not survive advection to these sites. Thus, despite the ongoing rapid Arctic warming, retreating sea-ice, and intruding Atlantic waters in the Eurasian Basin (Muilwijk et al., 2023), the perennial sea-ice that has remained in place currently still only permits one polar species to thrive: *N. pachyderma*. Indeed, net samplings conducted between 1985 and 2015 showed that subpolar species of foraminifera are not yet increasing in the region of the Fram Strait (Greco *et al.*, 2022). In fact, a decrease in subpolar species was found, which they linked to the increased export of Arctic sea-ice through this narrow gateway (Greco *et al.*, 2022). It was hypothesised that the invasion of the central Arctic Ocean by subpolar species will commence when ice export comes to a halt and the influence of Atlantic waters in the central Arctic increases (i.e. 'Atlantification' *sensu* Polyakov *et al.*, 2017). The dataset presented here provides an important baseline for comparison in future studies that will likely document this transformation. Of particular interest will be tracking the response *of N. pachyderma* as seasonal sea-ice disappears.

### 4.2. Spatial patterns of *N. pachyderma* abundance

In order determine their controlling variables, planktonic foraminifera abundances are commonly compared (correlated) with environmental parameters such as sea surface temperature (SST), sea surface salinity (SSS), chlorophyll *a*, and sea-ice cover (area coverage in %). In the case of the perennially ice-covered central Arctic Ocean (*SAS ODEN21* sites), it should be noted that both SST and SSS in the near-surface waters are strongly dictated by the ice pack, meaning that both SST and SSS were virtually constant across our study sites (respectively ca. -1.7°C and 30 g/kg). Therefore, neither SST nor SSS were able to explain the variability in standing stock of *N. pachyderma* across the sites in the central Arctic Ocean. Similarly, the position of the Atlantic water mass, as well as its maximum temperature (ca. 0.5°C), were extremely similar at six out of eight stations

(Figs. 4 and 5) indicating that they also do not contribute towards the observed variations in abundance across these sites. One consideration to make is that the spatial distribution of planktonic foraminifera populations within a given area is well known to exhibit 'patchiness' – meaning that populations are not distributed uniformly but can be characterised by marked differences in their abundance (e.g. Boltovskoy, 1971). In our survey, we found that the variability in abundance in the central Arctic Ocean ranged between 18-65 ind. m$^{-3}$ (average in top 200 m), indicating a degree of patchiness.

Multivariate statistics can be used to further explore the controlling factors. The results of the final Generalized Linear Model are summarized in Table 3. The analysis revealed significant coefficients ($p<0.05$) for the predictor variables salinity, temperature and distance to the sea-ice edge, with the coefficient for chlorophyll being insignificant. The largest coefficient was associated with salinity, suggesting it might represent one of the most the important controlling factors (negative

relationship with abundance). However, considering the small range of salinity variability in the 0-50 m (the depth interval where the highest abundances of foraminifera were found) between different sites and the limited size of the dataset, the analysis might suffer from overfitting and more data are needed to corroborate these preliminary findings.

The abundance of *N. pachyderma* (range 7.8-27.5 ind.m$^{-3}$; averages in the top 500 m) is comparable to that reported at the ice-

covered sites between 83°-86°N reported by Carstens and Wefer in 1992 (range 7.6-15.9 ind.m$^{-3}$; averages in the top 500m). Based on these two studies alone, it could perhaps be suggested that this range of abundances is typical for *N. pachyderma* under summer sea-ice, and that these have not markedly changed in the past 30 years. However, more studies are evidently needed to characterise both the spatial and temporal (seasonal/annual/decadal) trends of *N. pachyderma* abundance in the central Arctic Ocean. In order to put these numbers in a broader perspective, we compared our results with the abundance

reported near or outside of the seasonal ice edge, (re-)calculating the average in the top 200 m for these studies based on the original data (Table 2). The highest abundances of *N. pachyderma* have been observed in open waters located near the ice margin, where values were one magnitude higher than under sea ice (150-915 ind.m$^{-3}$; Carstens and Wefer, 1997; Table 2). In the North Atlantic Ocean, the lowest abundances observed along an east-west transect across the 75°N parallel (20 ind.m$^{-3}$) were comparable to those found under sea-ice, but maximum abundances were considerably higher (390 ind.m$^{-3}$; Stangeew,

2001). Overall, these observations probably reflect broad-scale spatial changes in primary productivity in the ocean, which is known to reach its highest values in the marginal ice zone (Carstens et al., 1997). Attempting to overcome the difference in mesh size with the study of Bé (1960), we use Berger's "equivalent catch" equation (Berger, 1969), which allows conversions between foraminiferal abundances obtained with different mesh sizes. Since the data of Bé is derived from irregular depth intervals, we selected those samples that differed by < 10m from the depth intervals used in our survey. Data were standardized

to a mesh size of 100 μm. For the 0-50 m depth interval, this conversion revealed standardised values ranging between 0.7-4.4 ind.m$^{-3}$ (median = 1.58 ind.m$^{-3}$) for the Bé study, compared to a range of 3.7-20.6 ind.m$^{-3}$ (median = 8.0 ind.m$^{-3}$) in our study. For the 0-100 m depth interval, two data points of 5.4 and 4.3 ind.m$^{-3}$ from the Bé study compared to standardised values ranging between 4.6-14.5 ind.m$^{-3}$ (median= 5.46 ind.m$^{-3}$) in our study. A plausible explanation for the elevated abundance

observed in our study is the difference in sampling timing, as Bé samples were collected in late September and October, likely when population numbers were declining.

In the region of the Lincoln Sea and adjoining fjords (*RYDER19*), abundances of planktonic foraminifera were extremely low, and are comparable to other studies reporting *N. pachyderma* abundance in shelf environments ($<2$ ind.m$^{-3}$; Kohfeld *et al.*, 1996; Zamelczyk *et al.*, 2021). It is indeed well-known that foraminiferal communities are rare in coastal and shelf environments (Schmuker 2000), especially in water depths $<100$ m. Common reasons to explain low abundances in (inner) shelf regions are the high variability in the physical and chemical environment, high turbidity and suspended sediment load, and shallow water depths impeding foraminifer reproduction cycles (Schmuker, 2000; Retailleau *et al.*, 2012; Zamelczyk *et al.*, 2021). Chlorophyll *a* concentrations are higher in the Lincoln Sea compared to Ryder Fjord, the latter exhibiting a narrower and deeper subsurface peak compared to the area outside the fjord (Fig. 10). This is consistent with the somewhat elevated foraminiferal abundances in the Lincoln Sea to outer Nares Strait region, compared to the very low abundances in Sherard Osborn Fjord (Fig. 6). However, we cannot discern the effect of primary productivity versus shallow bathymetry here. In the study of Retailleau *et al.* (2011), it was found that food availability (and freshwater input) were the main factors controlling standing stocks in the shallow Bay of Biscay, rather than depth itself. In contrast, Darling et al. (2007) showed an absence of planktonic foraminifera in the Bering Strait (generally shallower than 50 m), despite high productivity in the region. with similar temperatures and salinities.

### 4.3. Depth habitat of N. pachyderma

Our results confirm earlier studies, showing the shallow habitat of *N. pachyderma* underneath perennial sea ice (Bé, 1960; Carstens and Wefer, 1992; Volkmann, 2000). We speculate that the predominance of *N. pachyderma* in the upper 50 to 100 m in the central Arctic Ocean is related to food availability (presumably diatoms), which in turn depends on nutrient availability and light penetration limited by the presence of sea ice. Indeed, the chlorophyll *a* maximum is typically located between 20-40 m, corresponding to the depth interval where the maximum abundance of tests typically occurs (Fig. 5). A plausible explanation for the relatively high abundances found in the 50-100 m depth interval, located well below the chlorophyll maximum, could be the feeding of *N. pachyderma* on sinking aggregates, as proposed by the meta-analysis of Greco *et al.* (2019). The fact that the Polar Surface Water consists of cold and low salinity waters does not appear to hinder the resident *N. pachyderma* populations. These observations confirm previous suggestions that food availability/chlorophyll *a* concentration play a key role in determining the depth habitat of *N. pachyderma* (Kohfeld *et al.*, 1996; Volkmann, 2000; Pados & Spielhagen, 2014; Greco *et al.*, 2019) and that low salinity can be tolerated. This is further supported by the fact that highest abundances of *N. pachyderma* in the northern North Atlantic and Arctic Ocean ever reported were found in the highly productive marginal ice zone (Carstens and Wefer, 1997). Important to note is that a distinction should be made between the 'main depth habitat' of *N. pachyderma*, and the depth at which the secondary calcite is secreted, which is more important when it comes to interpreting geochemical signatures of fossil test in paleoceanography (Tell *et al.*, 2022). At all stations, test size increased

from the surface to 100-200 m depth, and at six out of eight stations no statistical difference was found between the 100-200 m and 200-500 m depth interval (Fig. 11; Fig. A4). This pattern would be consistent with reproduction taking place at the base of the productive zone, located at or below 100 m, and would provide some evidence that *N. pachyderma* does perform ontogenetic vertical migration (Manno & Pavlov, 2014; Tell *et al.*, 2022). On the other hand, large individuals were present at all depths, but rare (Fig. 11), perhaps substantiating that *N. pachyderma* performs both ontogenetic vertical migration as well as test growth at fixed depths, in line with the findings of Tell et al. (2022). Nevertheless, this result is to be confirmed and further analysed with a comprehensive analysis of *N. pachyderma* morphotypes in a follow-up study. To test whether the populations sampled at the different sites could represent a single reproductive cohort that had undergone synchronous reproduction, we plotted the mean size versus the date of sampling for the different depth intervals in the top 200 m (Fig. 10). The results showed there is no significant increase in test size with time, and thus point towards localised reproduction rather than a wider reproduction event effecting the whole region (Fig. 12).

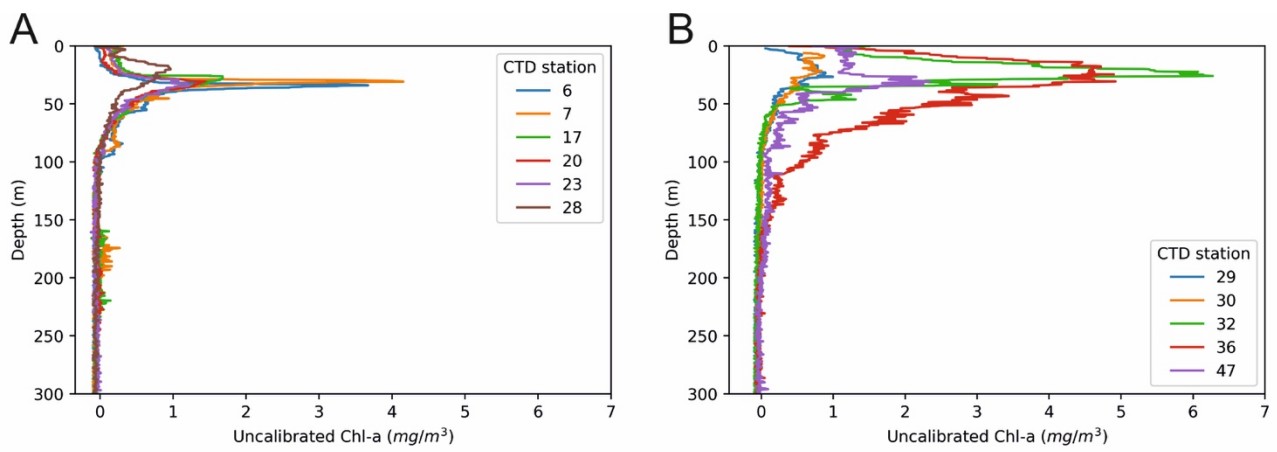

**Figure 10: Fluorescence-based estimate of chlorophyll *a* obtained during the *RYDER19* expedition. A) Profiles obtained within Sherard Osborn Fjord. B) Profiles obtained in the Lincoln Sea/Outer Nares Strait. Numbers indicate the names of the CTD stations, see Fig. 3 for their location.**

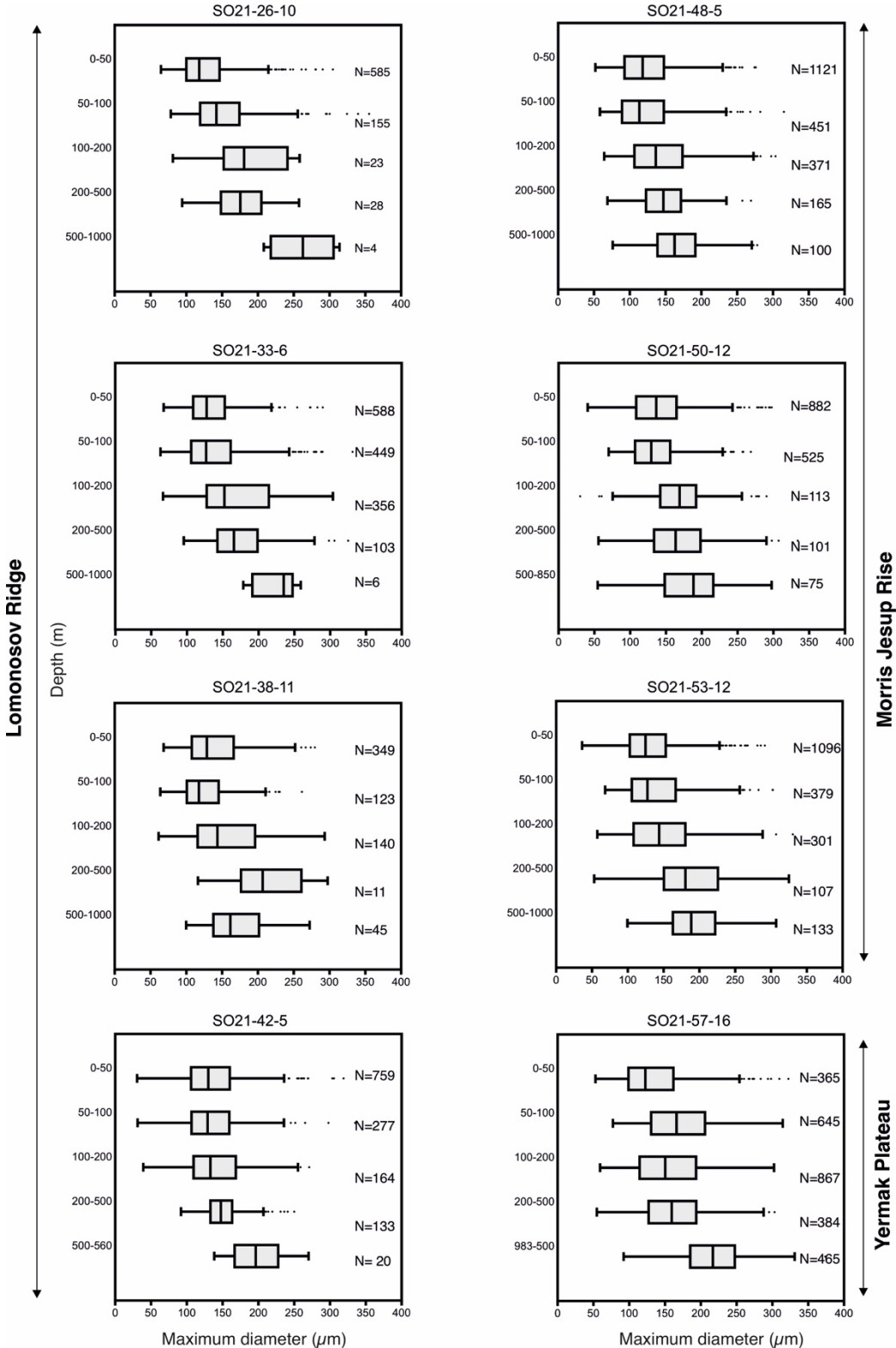

**Fig. 11. Size distributions of planktonic foraminiferal tests obtained at each station and depth interval (*SAS ODEN21*). Boxes delineate the interquartile range (IQR: Q1 to Q3), the line within the boxes represent the median. The upper and lower whiskers delineate Q3 + 1.5 IQR and Q1 – 1.5 IQR, respectively. Dots indicate outliers.**

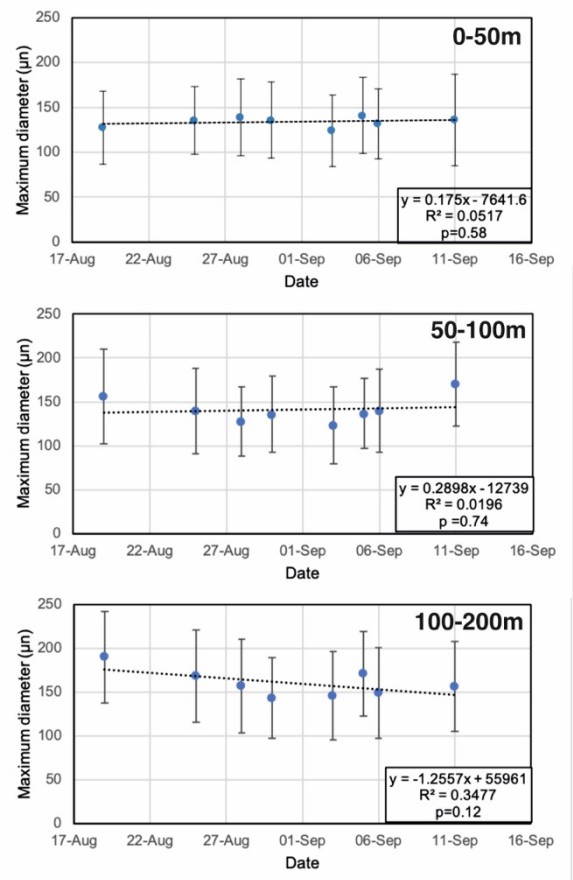

**Fig. 12. Maximum diameter of test size versus sampling date. Different panels indicate the results for the different depth intervals (0-50m, 50-100m, 100-200m). Whiskers denote one standard deviation. No significant trends were observed.**

## 5. Conclusion

This study details the first systematic survey of live planktonic foraminifera populations in the high Arctic Ocean, at sites near

the North Pole Area, southern Lomonosov Ridge, and the area north of Greenland. *Neogloboquadrina pachyderma* was the

only species present underneath the perennial ice cover in the region between the North Pole–Greenland at the time of

sampling, which would suggest that sub-polar species have not yet migrated into the central, perennially ice-covered Arctic Ocean. This is consistent with previous research showing that subpolar species are currently largely 'blocked' from entering the central Arctic basin due to increased sea-ice export through the Fram Strait (Greco et al., 2022), although another potential pathway exists via the Barents Sea. *Turborotalita quinqueloba* was only observed in very low numbers near the Yermak plateau and is absent at sites near the Lomonosov Ridge and the Lincoln Sea. This is consistent with its preference for Atlantic waters and abundance at/near the marginal ice zone, which has been widely reported. Overall, this observation emphasises the prominent oceanographic and climatic changes that must have occurred in the region of the central Arctic Ocean in the past, where evidence for a large-scale *T. quinqueloba* invasion during at least one previous interglacial is apparent (Vermassen et al., 2023).

Underneath perennial sea ice*, N. pachyderma* prefers a shallow habitat (in the top 50 to 100 m), in contrast to the ice marginal zone or areas with open water where it is more abundant at deeper water depths. We suggest that the shallow habitat is due to food availability, *i.e.* phytoplankton in the photic zone living at shallow depths under the sea-ice. The size distribution of N. pachyderma in the water column consistently revealed increasing test sizes with depth. At station SO21-26-10, cytoplasm content was categorised and counted, showing that >75% of the test were cytoplasm-bearing the upper 100 m, around 50% were cytoplasm-bearing between 200-500 m , and only empty tests were found below 500 m. This could perhaps represent a form of 'ontogenetic vertical migration', with individuals sinking as they grow, and reproducing around 100 m water depth. However, encrusted specimens were observed at all depths (albeit it at very low percentages in the top 100 m) and future studies deploying repeated tows would be needed to adequately determine the reproduction pattern of *N. pachyderma* under the summer sea ice.

As the Arctic Ocean is currently witnessing rapid environmental change, this dataset will provide an invaluable baseline for assessing the speed of change in the abundance and composition of planktonic foraminifera communities, in response to sea-ice decline and 'Atlantification', which are anticipated to intensify in the coming decades. Additionally, the study can serve as a valuable tool for enhancing palaeoceanographic investigations that use the sedimentary record.

Table 1. Multinet sampling stations during SAS Oden 2021.

| Expedition | Sampling station | Latitude | Longitude | Sampling depth (m) | Water depth (m) | Date | Time (UTC) | Type |
|---|---|---|---|---|---|---|---|---|
| SAS ODEN21 | SO21-26-10 | 89.126 | -150.593 | 1000 | 1341 | 2021-08-19 | 23:38:00 | Multinet |
| SAS ODEN21 | SO21-33-6 | 88.143 | -101.94 | 1000 | 2987 | 2021-08-25 | 16:27:00 | Multinet |
| SAS ODEN21 | SO21-38-11 | 87.747 | -66.488 | 1000 | 1180 | 2021-08-28 | 23:29:00 | Multinet |
| SAS ODEN21 | SO21-42-5 | 86.519 | -57.23 | 550 | 590 | 2021-08-30 | 23:09:00 | Multinet |
| SAS ODEN21 | SO21-48-5 | 84.927 | -33.51 | 1000 | 1539 | 2021-09-03 | 19:30:00 | Multinet |
| SAS ODEN21 | SO21-50-12 | 84.16 | -32.35 | 850 | 888 | 2021-09-05 | 01:22:00 | Multinet |
| SAS ODEN21 | SO21-53-4 | 84.462 | -23.99 | 975 | 1350 | 2021-09-06 | 00:17:00 | Multinet |
| SAS ODEN21 | SO21-58-16 | 82.37 | 8.485 | 983 | 983 | 2021-09-11 | 10:42:00 | Multinet |
| RYDER19 | Ryder19-01-PN | 82.344 | -59.817 | 300 | 440 | 2019-08-09 | N/A | Single net |
| RYDER19 | Ryder19-02-PN | 82.405 | -56.254 | 300 | 448 | 2019-08-10 | N/A | Single net |
| RYDER19 | Ryder19-03-PN | 82.024 | -52.144 | 300 | 837 | 2019-08-14 | N/A | Single net |
| RYDER19 | Ryder19-04-PN | 82 | -52.227 | 408 | 836 | 2019-08-15 | N/A | Single net |
| RYDER19 | Ryder19-05-PN | 81.884 | -50.988 | 250 | 267 | 2019-08-21 | N/A | Single net |
| RYDER19 | Ryder19-06-PN | 82.258 | -52.814 | 300 | 372 | 2019-08-25 | N/A | Single net |
| RYDER19 | Ryder19-07-PN | 82.477 | -54.218 | 300 | 485 | 2019-08-26 | N/A | Single net |
| RYDER19 | Ryder19-08-PN | 82.171 | -59.806 | 300 | 420 | 2019-08-31 | N/A | Single net |
| RYDER19 | Ryder19-09-PN | 81.64 | -64.271 | 300 | 623 | 2019-09-02 | N/A | Single net |
| RYDER19 | Ryder19-10-PN | 80.998 | -60.974 | 300 | 1043 | 2019-09-03 | N/A | Single net |

Table 2: Comparison of *N. pachyderma* abundance (ind.m$^{-3}$)across different sites in the northern North Atlantic

| Region | Sub region | Latitude | Top 200m range or average (ind.m-3) | Net size (µm) | Season | Ice conditons | Study |
|---|---|---|---|---|---|---|---|
| **Arctic Ocean (>80N)** | | | | | | | |
| | Lomonosov Ridge | 86.5-89N | 20-60 | >63 | Summer | ice-covered | This study |
| | Morris Jesup Rise | 84.5-85N | 20-65 | >63 | Summer | Ice-covered | This study |
| | Lincoln Sea/Sherard Osborn Fjord | 81-82N | 0.15-0.30 | >63 | Summer | locally open waters | This study |
| | Nansen Basin | 83-86N | 140 | >63 | Summer | ice-covered | Carstens & Wefer (1992) |
| | Nansen Basin | 81-83N | 140 | >63 | Summer | ice-covered | Carstens & Wefer (1992) |
| | Yermak Plateau | 82N | 30 | >63 | Summer | ice-covered | This study |
| **Fram Strait** | | | | | | | |
| | | 80N | 5-15 | >63 | Summer | ice-covered | Carstens & Wefer (1997) |
| | | 78N | 665 | >63 | Summer | ice margin | Carstens & Wefer (1997) |
| | | 78N | 150-915 | | Summer | open water | Carstens & Wefer (1997) |
| **Northeast Greenland** | | | | | | | |
| | NEW polynya | 80.5N | 0.25-20 | 150 | Summer | polynya | Kohfeld & Fairbanks (1996) |
| **Barents Sea** | | | | | | | |
| | | 76-82N | <2 | 63 | Winter | ice-covered | Zamelczyk et al. (2021) |
| **North Atlantic** | Nordic Seas | 75N | 20-390 | 63 | Summer | open water | Stangeew (2001) |

*Table 3. Summary statistics of the Generalised linear model*

| Parameter | Coefficient | Std. error | z-score | P>\|z\| | 2.5th percentile | 97.5th percentile |
|---|---|---|---|---|---|---|
| const | 3.5513 | 0.123 | 28.868 | 0 | 3.31 | 3.792 |
| avg_temperature | 0.3958 | 0.187 | 2.12 | 0.034 | 0.03 | 0.762 |
| avg_salinity | -0.9667 | 0.214 | -4.515 | 0 | -1.386 | -0.547 |
| chlorophyll | -0.0071 | 0.158 | -0.045 | 0.964 | -0.317 | 0.303 |
| distance_to_ice_edge | -0.2576 | 0.127 | -2.025 | 0.043 | -0.507 | -0.008 |

**Plate 1. SEM images illustrating the various morphotypes of *N. pachyderma*.**

1-3) Morphotype 'Nps-1'. 4-6). Morphotype 'Nps-2' 7-9). Morphotype 'Nps-3'. 10) Morphotype 'Nps-4'. 11-12) Morphotype 'Nps-5'. Specimens 1, 3, 5, 6, 7, and 8 are from the 200-100m depth interval at station SO21-58-16. Specimens 2, 9, and 10 are from the 850-500 m depth interval at SO21-50-12- specimen 4 is from the depth interval 200-100 m at SO21-50-12. Specimens 11 and 12 are from depth interval 50-0 m at station SO21-50-12. All scale bars are 100 μm, unless otherwise indicated.

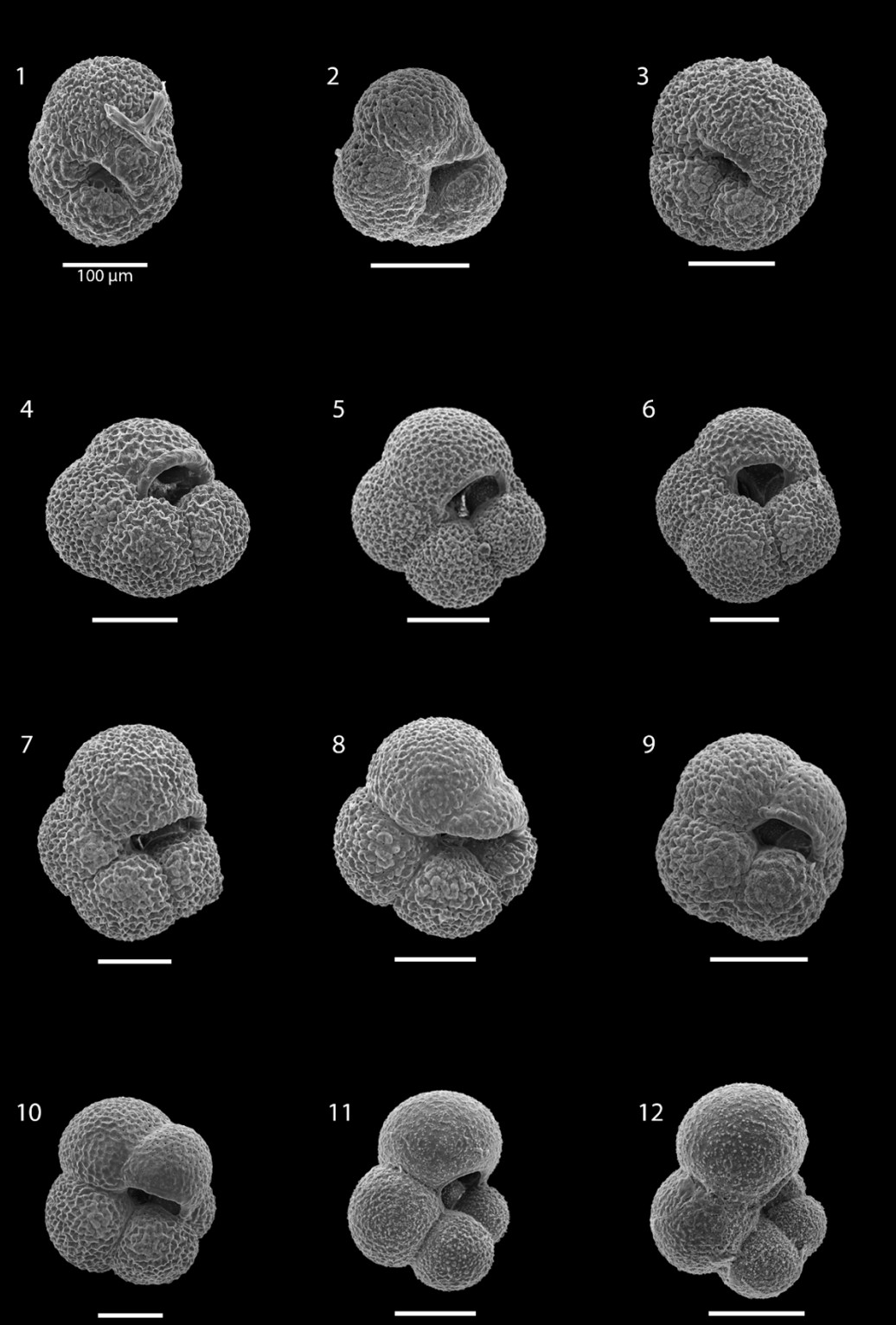

100 μm

**Plate 2 *Turborotalita quinqueloba***

Fig 1. Sample SO21-58-16, 500-200 m. Figs. 2, 4-11 Sample SO21-58-16, 200-100 m. Fig. 3 Wall texture of Figure 2. Fig. 6 Wall texture of Figure 5. Fig. 9 Wall texture of Figure 8. White arrows indicate spine holes, blue arrows indicate pores. All scale bars are 100 μm, unless otherwise indicated.

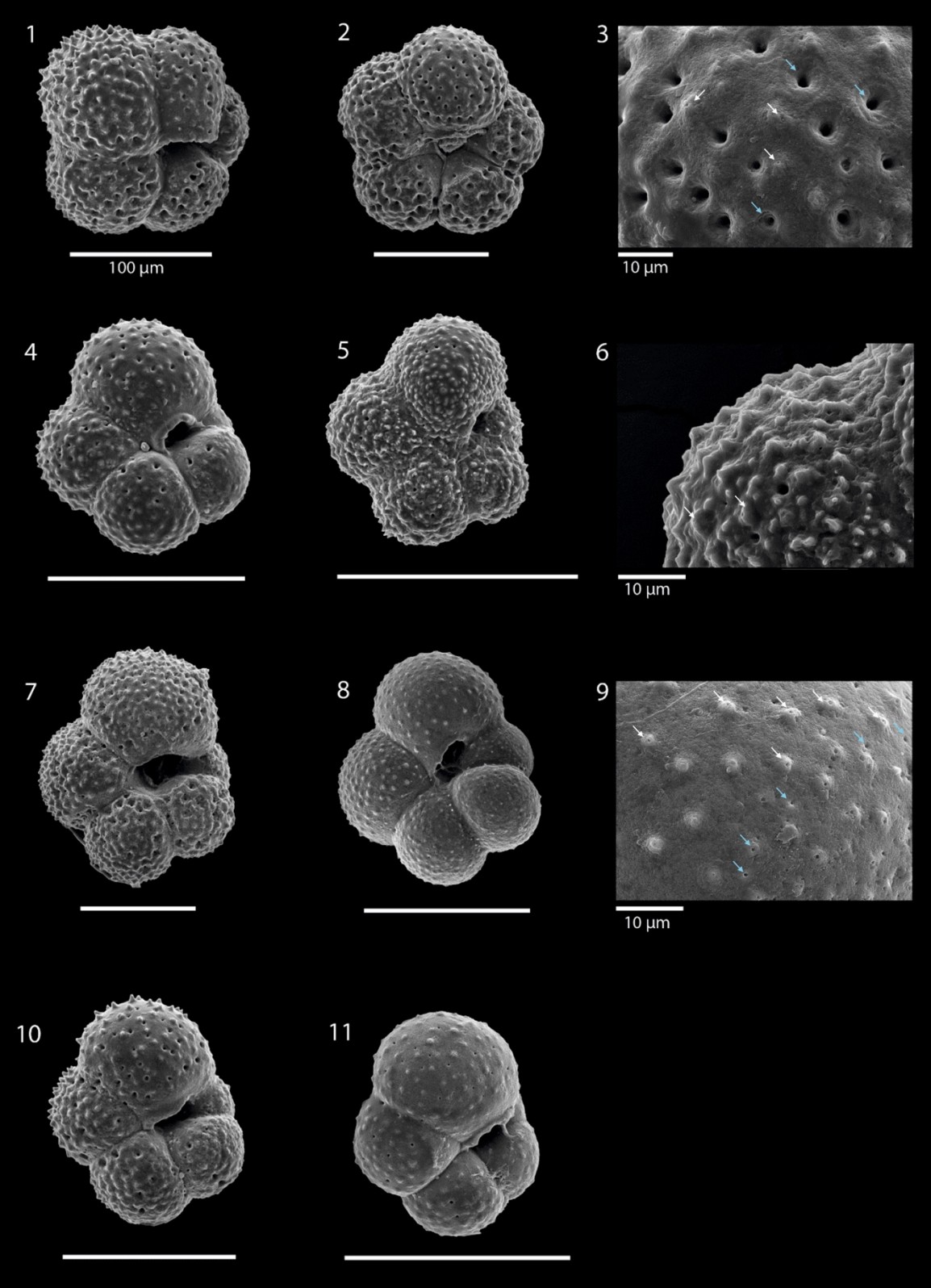

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

### Acknowledgements

We thank the crews and captains of IB Oden for skilfully operating the ship during both expeditions. The Polar Research Secretariat (SPRS) is thanked for assistance with handling of scientific equipment and expedition organisation. The scientific leadership of the expeditions SAD ODEN21 (Pauline Snoeijs-Leijonmalm) and RYDER19 (Martin Jakobsson, Larry Mayer) are thanked for planning and coordination of the expeditions.

### Funding

FV is supported by the Formas mobility grant for early-career researchers, grant nr 2022-02861. HC and FV acknowledge funding from VR grant DNR 2019-03757. CH and SK acknowledge funding from the Swedish Research Council (Vetenskapsrådet) Starting Research Grant (2018-03859). AH is supported by the Swedish Research Council (Vetenskapsrådet) Starting Research Grant (ÄR-NT 2020-03515).

### Author contributions

FV, CB, and HC designed the study. FV led the analysis and wrote the manuscript. FV and CB collected the foraminiferal data with the help of TW and AH. HF, CH, SK, CS, and MS collected and provided the environmental data. All authors contributed to the writing and revising of the manuscript.

### Data availability

The foraminiferal data related to this article is submitted and under revision at the Bolin Centre database to be made available for open access. CTD data is available at https://doi.pangaea.de/10.1594/PANGAEA.951266. Nutrient data is available at https://www.ncei.noaa.gov/access/metadata/landing-page/bin/iso?id=gov.noaa.nodc:0278647. Chlorophyll data is presented in Supplementary Table 1.

## 635 Competing interests

The authors have no competing interests to declare.

**Appendix**

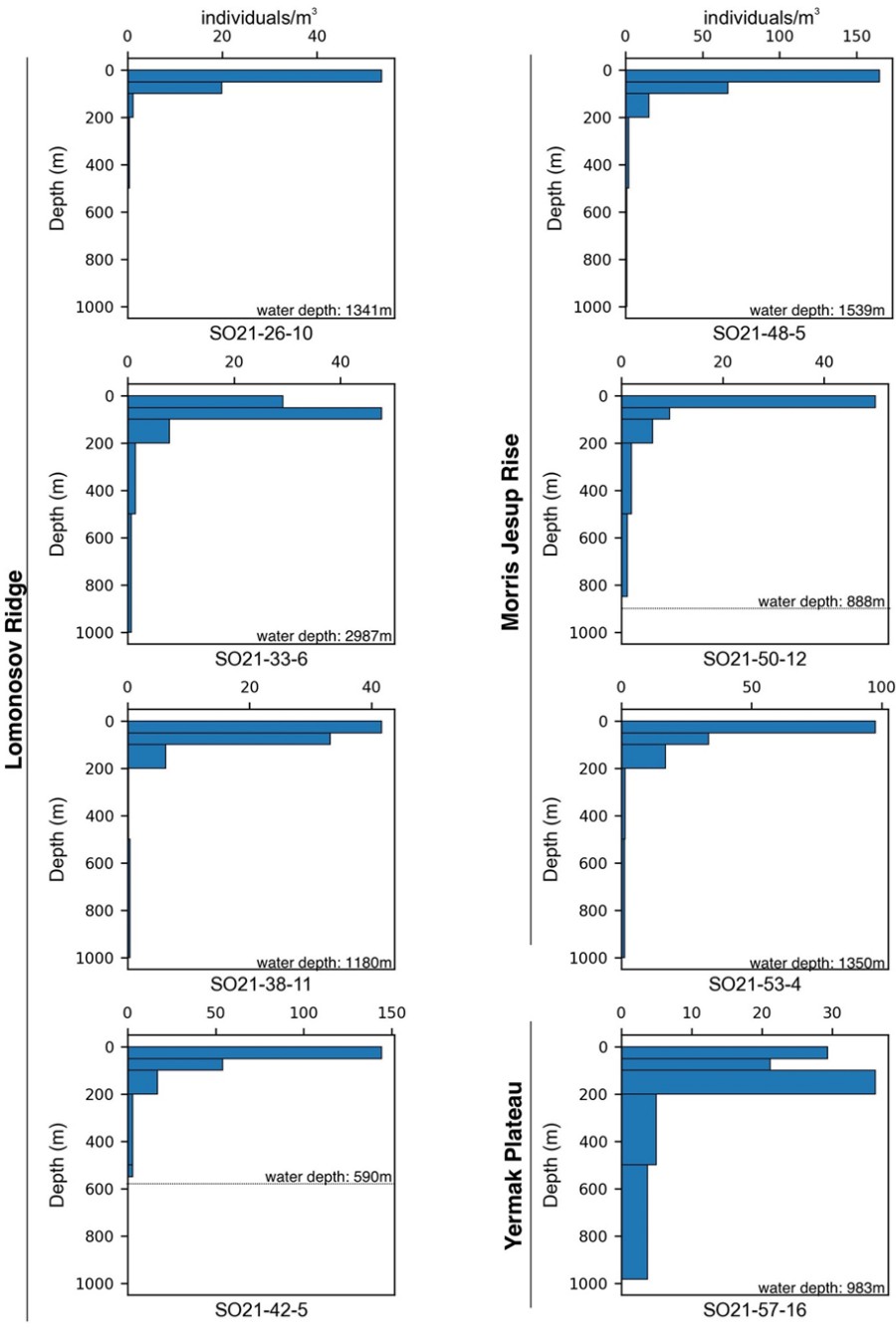

Figure A1. Planktonic foraminifer abundances at each site of *SAS ODEN* 21 (full depth down to 1000m)

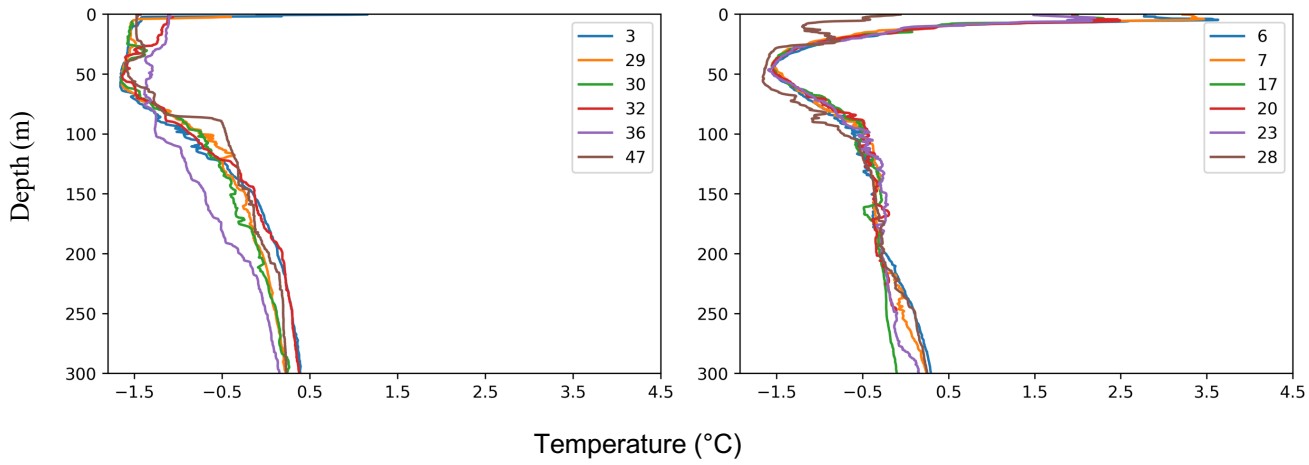

Figure A2: Temperature profiles in the Lincoln Sea and outer Nares Strait (Left) and Sherard Osborn Fjord (Right). Numbers indicate CTD stations.

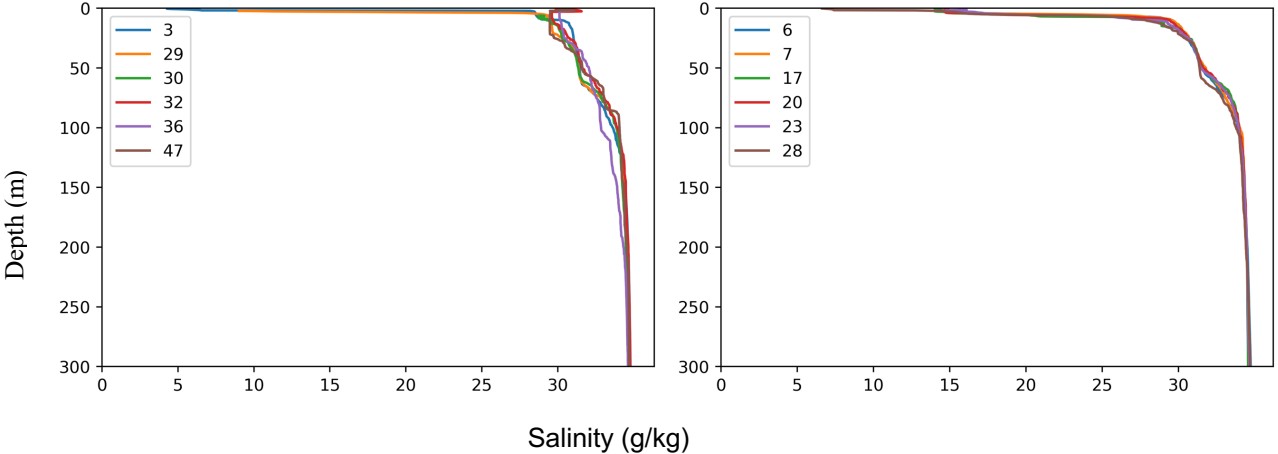

Figure A3: Salinity profiles fjord in the Lincoln Sea and Nares Strait (Left) and Sherard Osborn Fjord (Right)

**SO21-26-10**

| | 500-1000 | 200-500 | 100-200 | 50-100 | 0-50 |
|---|---|---|---|---|---|
| 500-1000 | | 0,0003304 | 0,007835 | 1,355E-06 | 0 |
| 200-500 | 5,893 | | 0,6696 | 0,3103 | 8,63E-07 |
| 100-200 | 4,721 | 1,888 | | 0,005512 | 0 |
| 50-100 | 7,52 | 2,708 | 4,865 | | 0 |
| 0-50 | 9,66 | 7,639 | 9,451 | 10,2 | |

**SO21-33-6**

| | 500-1000 | 200-500 | 100-200 | 50-100 | 0-50 |
|---|---|---|---|---|---|
| 500-1000 | | **0,1004** | 0,02613 | 8,19E-05 | 3,091E-05 |
| 200-500 | 3,481 | | **0,5153** | 2,681E-12 | 0 |
| 100-200 | 4,185 | 2,224 | | 8,511E-11 | 1,956E-12 |
| 50-100 | 6,333 | 10,44 | 9,771 | | **0,7305** |
| 0-50 | 6,629 | 11,11 | 10,49 | 1,747 | |

**SO21-38-11**

| | 500-1000 | 200-500 | 100-200 | 50-100 | 0-50 |
|---|---|---|---|---|---|
| 500-1000 | | **0,05563** | **0,4578** | 1,351E-06 | 0,000188 |
| 200-500 | 3,812 | | 0,001449 | 8,505E-08 | 2,827E-06 |
| 100-200 | 2,352 | 5,382 | | 2,563E-06 | 0,0007759 |
| 50-100 | 7,535 | 8,246 | 7,362 | | **0,1297** |
| 0-50 | 6,085 | 7,335 | 5,606 | 3,328 | |

**SO21-42-5**

| | 500-560 | 200-500 | 100-200 | 50-100 | 0-50 |
|---|---|---|---|---|---|

|  | 500-560 | 200-500 | 100-200 | 50-100 | 0-50 |
|---|---|---|---|---|---|
| 500-560 |  | 2,262E-07 | 6,712E-10 | 0 | 0 |
| 200-500 | 7,953 |  | 0,5258 | 0,008177 | 0,001704 |
| 100-200 | 9,305 | 2,201 |  | **0,4422** | **0,2939** |
| 50-100 | 10,78 | 4,696 | 2,386 |  | **1** |
| 0-50 | 11,12 | 5,307 | 2,751 | 0,02578 |  |

**SO21-48-5**

|  | 500-1000 | 200-500 | 100-200 | 50-100 | 0-50 |
|---|---|---|---|---|---|
| 500-1000 |  | 0,03626 | 0,001007 | 6,144E-11 | 6,145E-11 |
| 200-500 | 4,023 |  | **0,9236** | 4,671E-10 | 1,019E-10 |
| 100-200 | 5,491 | 1,164 |  | 6,224E-11 | 6,145E-11 |
| 50-100 | 12,34 | 9,384 | 10,63 |  | 0,9947 |
| 0-50 | 12,76 | 9,863 | 11,91 | 0,563 |  |

**SO21-50-12**

|  | 500-850 | 200-500 | 100-200 | 50-100 | 0-50 |
|---|---|---|---|---|---|
| 500-850 |  | 0,02385 | **0,06432** | 1,555E-12 | 1,554E-12 |
| 200-500 | 4,225 |  | **0,991** | 4,23E-10 | 2,447E-08 |
| 100-200 | 3,729 | 0,6467 |  | 2,172E-12 | 6,044E-11 |
| 50-100 | 13,48 | 9,39 | 10,69 |  | **0,4594** |
| 0-50 | 12,76 | 8,481 | 9,803 | 2,347 |  |

Figure A4: Results of Tukey's pairwise comparison, performed on the maximum diameter data, calculated for each
station. Numbers above the diagonal indicate p values, numbers below the diagonal indicate Tukey's Q value.
Numbers highlighted in yellow indicate pairs that are not statistically significant from each other.

Supplementary Table S1

| Station | Date | Latitude (DDM) | Longitude (DDM) | Depth (m) | Average Chl. Conc. (µg/L) |
|---|---|---|---|---|---|
| SO21-26-5 | 2021-08-19 | 89 6.674 | 150 0.584 | 10.000 | 0.129 |
| SO21-26-5 | 2021-08-19 | 89 6.674 | 150 0.584 | 30.000 | 0.263 |
| SO21-26-5 | 2021-08-19 | 89 6.674 | 150 0.584 | 40.000 | 0.155 |
| SO21-26-5 | 2021-08-19 | 89 6.674 | 150 0.584 | 50.000 | 0.140 |
| SO21-26-5 | 2021-08-19 | 89 6.674 | 150 0.584 | 75.000 | 0.051 |
| SO21-26-5 | 2021-08-19 | 89 6.674 | 150 0.584 | 100.000 | 0.017 |
| SO21-26-5 | 2021-08-19 | 89 6.674 | 150 0.584 | 125.000 | 0.011 |
| SO21-26-5 | 2021-08-19 | 89 6.674 | 150 0.584 | 150.000 | 0.011 |
| SO21-26-5 | 2021-08-19 | 89 6.674 | 150 0.584 | 200.000 | 0.009 |
| SO21-26-5 | 2021-08-19 | 89 6.674 | 150 0.584 | 250.000 | 0.007 |
| SO21-26-5 | 2021-08-19 | 89 6.674 | 150 0.584 | 500.000 | 0.003 |

| Station | Date | Latitude (DDM) | Longitude (DDM) | Depth (m) | Average Chl. Conc (µg/L) |
|---|---|---|---|---|---|
| SO21-33-5 | 2021-08-25 | 88 8.329 | 102 0.811 | 10.000 | 0.078 |
| SO21-33-5 | 2021-08-25 | 88 8.329 | 102 0.811 | 30.000 | 0.181 |
| SO21-33-5 | 2021-08-25 | 88 8.329 | 102 0.811 | 40.000 | 0.159 |
| SO21-33-5 | 2021-08-25 | 88 8.329 | 102 0.811 | 43.000 | 0.175 |
| SO21-33-5 | 2021-08-25 | 88 8.329 | 102 0.811 | 50.000 | 0.132 |
| SO21-33-5 | 2021-08-25 | 88 8.329 | 102 0.811 | 75.000 | 0.036 |
| SO21-33-5 | 2021-08-25 | 88 8.329 | 102 0.811 | 100.000 | 0.007 |
| SO21-33-5 | 2021-08-25 | 88 8.329 | 102 0.811 | 125.000 | 0.007 |
| SO21-33-5 | 2021-08-25 | 88 8.329 | 102 0.811 | 150.000 | 0.005 |
| SO21-33-5 | 2021-08-25 | 88 8.329 | 102 0.811 | 200.000 | 0.006 |
| SO21-33-5 | 2021-08-25 | 88 8.329 | 102 0.811 | 370.000 | 0.002 |
| SO21-33-5 | 2021-08-25 | 88 8.329 | 102 0.811 | 500.000 | 0.001 |

| Station | Date | Latitude (DDM) | Longitude (DDM) | Depth (m) | Average Chl. Conc (µg/L) |
|---|---|---|---|---|---|
| SO21-38-17 | 2021-08-29 | 87 46.54 | 65 49.43 | 10.000 | 0.152 |
| SO21-38-17 | 2021-08-29 | 87 46.54 | 65 49.43 | 30.000 | 0.246 |
| SO21-38-17 | 2021-08-29 | 87 46.54 | 65 49.43 | 31.400 | 0.237 |
| SO21-38-17 | 2021-08-29 | 87 46.54 | 65 49.43 | 40.000 | 0.201 |
| SO21-38-17 | 2021-08-29 | 87 46.54 | 65 49.43 | 50.000 | 0.095 |
| SO21-38-17 | 2021-08-29 | 87 46.54 | 65 49.43 | 75.000 | 0.014 |

| Station | Date | Latitude (DDM) | Longitude (DDM) | Depth (m) | Average Chl. Conc (µg/L) |
|---|---|---|---|---|---|
| SO21-38-17 | 2021-08-29 | 87 46.54 | 65 49.43 | 100.000 | 0.008 |
| SO21-38-17 | 2021-08-29 | 87 46.54 | 65 49.43 | 125.000 | 0.011 |
| SO21-38-17 | 2021-08-29 | 87 46.54 | 65 49.43 | 150.000 | 0.009 |
| SO21-38-17 | 2021-08-29 | 87 46.54 | 65 49.43 | 200.000 | 0.004 |
| SO21-38-17 | 2021-08-29 | 87 46.54 | 65 49.43 | 380.000 | 0.002 |
| SO21-38-17 | 2021-08-29 | 87 46.54 | 65 49.43 | 500.000 | 0.001 |

| Station | Date | Latitude (DDM) | Longitude (DDM) | Depth (m) | Average Chl. Conc (µg/L) |
|---|---|---|---|---|---|
| SO21-42-8 | 2021-08-31 | 86 31.248 | 57 6.016 | 10.000 | 0.086 |
| SO21-42-8 | 2021-08-31 | 86 31.248 | 57 6.016 | 30.000 | 0.174 |
| SO21-42-8 | 2021-08-31 | 86 31.248 | 57 6.016 | 40.000 | 0.249 |
| SO21-42-8 | 2021-08-31 | 86 31.248 | 57 6.016 | 45.000 | 0.298 |
| SO21-42-8 | 2021-08-31 | 86 31.248 | 57 6.016 | 50.000 | 0.185 |
| SO21-42-8 | 2021-08-31 | 86 31.248 | 57 6.016 | 75.000 | 0.031 |
| SO21-42-8 | 2021-08-31 | 86 31.248 | 57 6.016 | 100.000 | 0.016 |
| SO21-42-8 | 2021-08-31 | 86 31.248 | 57 6.016 | 125.000 | 0.007 |
| SO21-42-8 | 2021-08-31 | 86 31.248 | 57 6.016 | 150.000 | 0.007 |
| SO21-42-8 | 2021-08-31 | 86 31.248 | 57 6.016 | 200.000 | 0.004 |
| SO21-42-8 | 2021-08-31 | 86 31.248 | 57 6.016 | 320.000 | 0.006 |
| SO21-42-8 | 2021-08-31 | 86 31.248 | 57 6.016 | 500.000 | 0.001 |

| Station | Date | Latitude (DDM) | Longitude (DDM) | Depth (m) | Average Chl. Conc (µg/L) |
|---|---|---|---|---|---|
| SO21-48-4 | 2021-09-03 | 84 55.495 | 33 28.730 | 10.000 | 0.125 |
| SO21-48-4 | 2021-09-03 | 84 55.495 | 33 28.730 | 30.000 | 0.418 |
| SO21-48-4 | 2021-09-03 | 84 55.495 | 33 28.730 | 40.000 | 0.087 |
| SO21-48-4 | 2021-09-03 | 84 55.495 | 33 28.730 | 50.000 | 0.026 |
| SO21-48-4 | 2021-09-03 | 84 55.495 | 33 28.730 | 75.000 | 0.008 |
| SO21-48-4 | 2021-09-03 | 84 55.495 | 33 28.730 | 100.000 | 0.008 |
| SO21-48-4 | 2021-09-03 | 84 55.495 | 33 28.730 | 125.000 | 0.008 |
| SO21-48-4 | 2021-09-03 | 84 55.495 | 33 28.730 | 150.000 | 0.006 |
| SO21-48-4 | 2021-09-03 | 84 55.495 | 33 28.730 | 200.000 | 0.006 |
| SO21-48-4 | 2021-09-03 | 84 55.495 | 33 28.730 | 335.000 | 0.002 |
| SO21-48-4 | 2021-09-03 | 84 55.495 | 33 28.730 | 500.000 | 0.003 |

| Station | Date | Latitude (DDM) | Longitude (DDM) | Depth (m) | Average Chl. Conc (µg/L) |
|---|---|---|---|---|---|
| SO21-50-13 | 2021-09-04 | 84 9.565 | 32 21.380 | 10.000 | 0.224 |
| SO21-50-13 | 2021-09-04 | 84 9.565 | 32 21.380 | 27.000 | 1.057 |
| SO21-50-13 | 2021-09-04 | 84 9.565 | 32 21.380 | 30.000 | 0.548 |
| SO21-50-13 | 2021-09-04 | 84 9.565 | 32 21.380 | 40.000 | 0.383 |

| Station | Date | Latitude (DDM) | Longitude (DDM) | Depth (m) | Average Chl. Conc (µg/L) |
|---|---|---|---|---|---|
| SO21-50-13 | 2021-09-04 | 84 9.565 | 32 21.380 | 50.000 | 0.166 |
| SO21-50-13 | 2021-09-04 | 84 9.565 | 32 21.380 | 75.000 | 0.046 |
| SO21-50-13 | 2021-09-04 | 84 9.565 | 32 21.380 | 100.000 | 0.015 |
| SO21-50-13 | 2021-09-04 | 84 9.565 | 32 21.380 | 125.000 | 0.020 |
| SO21-50-13 | 2021-09-04 | 84 9.565 | 32 21.380 | 150.000 | 0.016 |
| SO21-50-13 | 2021-09-04 | 84 9.565 | 32 21.380 | 200.000 | 0.013 |
| SO21-50-13 | 2021-09-04 | 84 9.565 | 32 21.380 | 350.000 | 0.006 |
| SO21-50-13 | 2021-09-04 | 84 9.565 | 32 21.380 | 500.000 | 0.002 |

| Station | Date | Latitude (DDM) | Longitude (DDM) | Depth (m) | Average Chl. Conc (µg/L) |
|---|---|---|---|---|---|
| SO21-53-9 | 2021-09-06 | 84 31.308 | 24 24.236 | 10.000 | 0.059 |
| SO21-53-9 | 2021-09-06 | 84 31.308 | 24 24.236 | 30.000 | 0.115 |
| SO21-53-9 | 2021-09-06 | 84 31.308 | 24 24.236 | 32.000 | 0.118 |
| SO21-53-9 | 2021-09-06 | 84 31.308 | 24 24.236 | 40.000 | 0.093 |
| SO21-53-9 | 2021-09-06 | 84 31.308 | 24 24.236 | 50.000 | 0.062 |
| SO21-53-9 | 2021-09-06 | 84 31.308 | 24 24.236 | 75.000 | 0.016 |
| SO21-53-9 | 2021-09-06 | 84 31.308 | 24 24.236 | 100.000 | 0.006 |
| SO21-53-9 | 2021-09-06 | 84 31.308 | 24 24.236 | 125.000 | 0.006 |
| SO21-53-9 | 2021-09-06 | 84 31.308 | 24 24.236 | 150.000 | 0.004 |
| SO21-53-9 | 2021-09-06 | 84 31.308 | 24 24.236 | 200.000 | 0.005 |
| SO21-53-9 | 2021-09-06 | 84 31.308 | 24 24.236 | 355.000 | 0.002 |
| SO21-53-9 | 2021-09-06 | 84 31.308 | 24 24.236 | 500.000 | 0.002 |

| Station | Date | Latitude (DDM) | Longitude (DDM) | Depth (m) | Average Chl. Conc (µg/L) |
|---|---|---|---|---|---|
| SO21-57-12 | 2021-09-11 | 82 27.907 | 8 41.821 | 10.000 | 0.446 |
| SO21-57-12 | 2021-09-11 | 82 27.907 | 8 41.821 | 15.000 | 0.438 |
| SO21-57-12 | 2021-09-11 | 82 27.907 | 8 41.821 | 30.000 | 0.299 |
| SO21-57-12 | 2021-09-11 | 82 27.907 | 8 41.821 | 40.000 | 0.158 |
| SO21-57-12 | 2021-09-11 | 82 27.907 | 8 41.821 | 50.000 | 0.170 |
| SO21-57-12 | 2021-09-11 | 82 27.907 | 8 41.821 | 75.000 | 0.059 |
| SO21-57-12 | 2021-09-11 | 82 27.907 | 8 41.821 | 100.000 | 0.051 |
| SO21-57-12 | 2021-09-11 | 82 27.907 | 8 41.821 | 125.000 | 0.027 |
| SO21-57-12 | 2021-09-11 | 82 27.907 | 8 41.821 | 150.000 | 0.022 |
| SO21-57-12 | 2021-09-11 | 82 27.907 | 8 41.821 | 200.000 | 0.012 |
| SO21-57-12 | 2021-09-11 | 82 27.907 | 8 41.821 | 270.000 | 0.009 |
| SO21-57-12 | 2021-09-11 | 82 27.907 | 8 41.821 | 500.000 | 0.006 |