# Peer review of "The distribution and abundance of planktonic foraminifera under summer sea-ice in the Arctic Ocean"

_EGUsphere, 2024_

## Author Response (AR1)

**Response to Reviewer 1**

*I reviewed the manuscript of Dr. Flor Vermassen* et al. *and think it is an important contribution to our scientific community. The data presented in the manuscript are rather rare and provide, as written by the authors themselves, an interesting baseline for assessing the speed of change in the abundance and composition of planktonic foraminifera assemblages in the region for the coming decades.*

*I therefore support the acceptance of this manuscript after some rather minor revisions.*

> We thank the referee for a careful and useful review which points out a series of (minor) corrections. We agree that the way in which we quantified and interpreted cytoplasm-bearing versus empty test should be clarified and now have done so (see below).

*While the study is interesting and provides unique data, it mostly confirms previous findings and I believe that the authors could go a bit deeper in some of their observations and discussion, especially about the distinction of "living" and "dead" specimens.*

*It is unclear if the authors systematically distinguished cytoplasm-bearing and empty specimens. This, however, is important as they refer to "living foraminifera" often in the manuscript and consider that all specimens sampled were "living" rather than transported there. A distinction was made (mentioned in the conclusion as well as images of specimens with cytoplasm) but it is not numerically discussed or displayed in the manuscript and thus, leaves a lot of unknown concerning what would (or not) support their conclusions.*

> We agree that we had not been fully clear whether a systematic distinction was made between cytoplasm-bearing and empty specimens. The answer is that a systematic counting of cytoplasm content was only performed at one station (SO21-26-10). Our observations while picking the specimens at other stations, although not quantitative, confirmed the general pattern found in SO21-26-10, namely that the bigger /thicker shells at greater depths were indeed generally empty. Vice versa, we observed that the thinner, smaller shells in the shallower samples were virtually all cytoplasm bearing. We have now added a new panel to Fig. 9, showing the cytoplasm percentages (Panel 9B). In addition, we made clear that we have assumed that cytoplasm-bearing individuals were alive, i.e. 'living', and that the empty tests were presumed to be dead and sinking. We also clarified the methodology in lines 139-145.

*Another point: The authors sampled relatively close to the coasts in or near fjords and in very shallow environments (several stations with a bathymetry <500 m depth). It has already been discussed that planktonic foraminifera do not leave/are not abundant in such environments (e.g., Schmuker 2000). I believe that this should appear somewhere in the discussion.*

We added a sentence in the discussion that points this out in a more straightforward manner (Line 581) :

*Specific comments:*

*Introduction:*

*L. 55-57, which month were the plankton tows sampled for these studies?*

Pados: Late June/early July 2011

Volkmann: July/August

Carstens: August

Darling: July

*L. 60, and G. uvula, displaying similarities in their ecological preferences with T. quinqueloba.*

We feel this addition doesn't fit the sentence structure.

*L. 65, the species is not resident but it could still stay there for months and reproduce there – we do not know.*

Yes, we agree.

*Figure 2: a) It is difficult to read the legend, written in white on a light grey background. Please write it in black and increase the text size. b) please provide the station location.*

We improved the fig/legend for legibility. The station locations are provided in three other figures, adding them here would obscure the satellite image.

*Methods:*

*L. 107, a multinet is generally equipped with 5 nets, why did the authors sample only 4 depths intervals, also, why sampling down to 1000 m depths and not with the classical 700 to 0 m?*

The interval 200-100m was missing in the text, we added this. We were not aware 700m was more classical than 700m.

*L. 108, why stop the net between intervals?*

To make sure the net closed before hauling the next section.

*L. 111, for samples picked in Sweden, was the pH measured beforehand?*

No, but pteropods were still plentiful and we did not observe signs of dissolution.

*2.2, were all specimens manually oriented the same way on slides? In other words, are the size measurements of each foraminifera comparable? If not, what is the error linked to particles/foraminifera orientation?*

Specimens were either placed on their dorsal or ventral side (but not sideways). Therefore, measurements of surface area and maximal diameter are comparable.

*Overall, was a distinction made between living/cytoplasm-bearing and dead/empty specimens? If yes, please provide details.*

Counting of cytoplasm content (colour) was only performed at one station (SO21-26-10), as mentioned above.  At other stations, we were unable to count cytoplasm colour variants due to time constraints, but visual observations confirmed the general pattern observed at station SO21-26-10- (See also response to the first comment). See the new lines 139-145.

*Results:*

*L. 183 – 187, which water masses are present between 150 and 500m depth?*

We had phrased it awkwardly and have rewritten the sentence.

*Figure 3: What is Figure X?*

Figure 4, we fixed this.

*3.2.1. Are the concentrations presented here the ones of all foraminifera living and dead (cytoplasm bearing and empty shells)?*

Yes. We have now clarified this in the methods.

"The reported concentrations include all tests, regardless of cytoplasm content."

*L. 262, Living in the top 100m or found/present in the top 100m?*

It is indeed more correct to state "present" and we changed this.

*L. 265, same comment.*

Changed to "present"

*L. 284, the authors found more (relative) T. quinqueloba in depth, from 200 to 1000m depth. Could they discuss/comment this finding in the manuscript?*

The numbers are a bit too thin to be firm about this, perhaps they were being advected with the Atlantic Water.

*L. 295, what does "predominant" mean? Could you give a relative percentage of cytoplasm-bearing specimens at the different depths?*

Absolutely, we have now done this in the new Fig 9B and added the numbers, see the new paragraph 418-423.

*Figure 8: Specimen c has a "decaying" cytoplasm suggesting the individual is probably dying. Based on the histogram (left) I do not think that there is (statistically) relatively more "red" specimens in the surface layer than deeper. Maybe the authors could comment on that and specify if it is significant or not. finally, for this figure, the cytoplasm color is not discussed in the manuscript, I am not sure it provides very relevant information.*

There is a 20% difference between the 0-50 and 50-100m interval at station SO21-26, which we now state in the manuscript (see previous comment). We have deliberately not included the potential causes of the cytoplasm colours in this manuscript, as it will be the topic of a follow-up paper, focusing on the microbiome. We thought it preferable to introduce it in this manuscript, as it was a notable occurrence.

*Discussion:*

*L. 310, without a clear distinction of cytoplasm-bearing / empty test, it is difficult to be so adamant in the wording. Even if I agree, I would suggest the authors to slightly tone down.*

With the new Fig. 9B and clarifications made regarding cytoplasm (see previous comments), this statement should now be better supported.

*L. 327-328, which "could" suggest that... I would add the could. Sampling, picking, storage, etc., usually break the spines, even in very healthy specimens.*

Yes, we definitely agree and have made the change

*L. 357, please also provide the p-value*

We changed the statistical approach according to reviewer 3's comments.

*L. 366 to 374, please provide seasons or month(s) for the number from the literature mentioned. It is indeed well-known that the species displays a highly seasonal pattern of abundances.*

Season of sampling is noted in Table 2. We now also note that Bé (1960) survey was conducted during late September/October.

*L. 378-379, planktonic foraminifera are also just very well-known to not inhabit such environments... Rather than food availability only, what could explain the low concentrations is just the sampling sites and the bathymetry.*

Yes, see new line 584.

*L. 405, also cite Manno and Pavlov 2014*

OK, done

*L. 405 – 408, the authors should here again stress the fact that no clear distinction between cytoplasm-bearing and empty shells was made. Are the bigger and/or thicker shells found more in depth systematically empty? Do they also observe thinner, smaller empty shells in the shallow samples (suggesting a reproductive event without gametogenesis...)?*

Quantification of cytoplasm colour was done at one station, see the new Fig 9B. Indeed, the reviewer correctly points out that bigger and/or thicker shells found at the deepest depths were systematically empty (see the 500-1000m interval at that station). We have a follow-up study in the pipeline which will discuss *N. pachyderma* reproduction (also including coiling data).

*Conclusions:*

*L. 431-435, the authors should discuss the bathymetry and the fact that some stations (from what I understood) are rather close to the shore.*

Agreed, see the new amendments made regarding the role of bathymetry and coastal environment in the discussion ( lines 581-594).

*L. 435-436, are cytoplasm individuals dominating the surface samples? No data in the manuscript show that. This is very likely true and I agree but, one should provide evidence (numerical) for that.*

We agree that we had not been not clear about this. We have now added an additional figure showing that cytoplasm individuals are more abundant in the surface level at station SO21-26-10 – see the new Fig. 9B and we have added the numbers to the text.

**Response to Reviewer 2**

*The manuscript of Vermassen and coauthors on 'The distribution and abundance of planktonic foraminifera under summer sea-ice in the Arctic Ocean' presents a new and unique data set on the understanding of Neogloboquadrina pachyderma from so far uncharted regions between Greenland and the North Pole, and confirms earlier findings on the population dynamics, and which would need to be discussed in more depth.*

We thank the reviewer for their thorough review and useful suggestions, which will substantially improve the manuscript. We have resolved the concern regarding the main conclusion, by being clearer about which specific regions our interpretations are valid – i.e. the perennially ice-covered Arctic Ocean, not the entire Arctic Ocean.

*However, I don't concur with the major conclusion of the manuscript that 'N. pachyderma is the only species present underneath the perennial ice cover in the region between the North Pole–Greenland, and that sub-polar species have not migrated into the central Arctic Ocean', and which should be tuned down and changed to avoid overinterpretation.*

We can see now that our text appeared to suggest that no species are migrating at all in the *entire* Arctic Ocean. We now clarify that our results and interpretations are valid only for the *perennially ice-covered part* of the central Arctic Ocean.

In the abstract, we write

"Our results would suggest that the anticipated turnover from polar to subpolar planktonic species in the **perennially ice-covered part** of the central Arctic Ocean has not yet occurred, in agreement with a recent meta-analysis from the Fram Strait which suggested that increased export of sea-ice is blocking the influx of Atlantic-sourced species."

In the conclusions, we write

"*N. pachyderma* was the only species present underneath the perennial ice cover in the region between the North Pole–Greenland **at the time of sampling**, which would suggest that sub-polar species have not yet migrated into the **central, perennially ice-covered Arctic Ocean**. This is consistent with previous research showing that subpolar species are currently largely 'blocked' from entering the central Arctic basin due to increased sea-ice export through the Fram Strait (Greco et al., 2022), **although another potential pathway exists via the Barents Sea**"

*Plankton net samples from several sampling locations only document a snapshot of the population dynamics of a large region, and cannot conclude on missing elements (other planktic foraminifer species) and on the entire Arctic Ocean.*

We agree that plankton net samples only document a snapshot of the population dynamics of a large region and species can be 'missed'. Nevertheless, our data, taken over the course of 6 weeks, were consistent in terms of species composition.  We would like to clarify that we certainly did not intend to speculate on the status of the 'entire' Arctic Ocean – our data and interpretation concern the perennially ice-covered part of the central Arctic Ocean. See the changed text in relation the previous comment.

*Also, the rather complex current system of the Arctic Ocean would need to be analyzed and interpreted for the transport of Atlantic plankton elements to know when and where any of these may arrive. For example, in the Abstract it's stated that the '… results indicate that the anticipated turnover from polar to subpolar planktonic species in the Arctic Ocean has not yet occurred, in agreement with recent studies from the Fram Strait', and which is based on the misunderstanding of the Artic circulation pattern; the Fram Strait (as well as the sampling sites north of Greenland) are located at the 'end' of the Arctic circulation where Arctic waters are transported into the GIN seas and the North Atlantic. Therefore, it's very unlikely to find Atlantic sourced plankton north of Greenland and in the Fram Strait.*

We are a bit surprised with the statement that the entire Fram Strait would *only* receive Atlantic Waters that have entirely recirculated around the Arctic Ocean.  The well-known West Spitsbergen Current runs through the Fram Strait and provides a direct pathway of Atlantic Waters into the central Arctic Ocean, providing a plausible pathway for Atlantic (subpolar) species to enter the central Arctic Ocean. Several studies have previously reported Atlantic-sourced species in the Fram Strait (Carstens & Wefer (1997); Pados & Spielhagen (2014); Volkmann (2000)) . We have clarified the role of the West Spitsbergen Current throughout the manuscript, e.g. lines 69-71.

*New papers are accepted for publication and in print, which show the invasion of new species in polar waters from the North Atlantic Current entering the Arctic Ocean on the eastern (Norwegian and Russian) side, which has not been sampled here. Therefore, it may be wise not to make too general statements on this topic.*

We agree and clarified in the manuscript that our data and interpretations concern the *perennially ice-covered part* of the central Arctic Ocean. We believe our statements are therefore certainly compatible with subpolar species entering Norwegian and Russian shelves, which are located in the seasonal ice zone and indeed along the pathway of incoming Atlantic Water,

as has also been reported previously by Volkmann (2000). It makes sense that these seasonally ice-free environments would be invaded first, before the more central parts of the perennially ice-covered Arctic Ocean.

*Overall, the paper reads good, but includes many little flaws, which need to be corrected before the paper can be accepted for publications. For example, salinity has no unit; all maps would need coordinates (N, E, and W) for orientation (e.g., Figs. 1 and 2); I guess I know what the Figure 7 should show, but the information is difficult to from the rather 'special' kind of design, and the authors may convey the information in a clearer kind of way. 'Concentration' of foraminifers is seawater needs to be changed to 'standing stock'.*

We apologise for the little flaws and thank the reviewer for pointing these out.  We have made the corrections accordingly. We agree that Fig.7 is somewhat non-traditional, but it appears this style of visualisation is used more and more, see e.g. Snoeijs-Leijonmalm et al., 2022, *Sci. Adv*. For the salinity unit we follow the latest international standards (TEOS10) and quote values of absolute salinity, in g/kg.

*Page 2, lines 49-50: Please give references for the 'knowledge of resident pelagic communities in the remote, perennially ice-covered regions is minimal.' There are quite some papers available on planktic forams and other sea-ice related biota. For N. pachyderma and T. quinqueloba, please also have a look at the paper of Simstich et al. (2003, ecology and isotopes).*

For the specific region we wanted to discuss here the (the *perennially ice-covered* Arctic regio), there are only two studies on live planktonic foraminifera (Bé 1960 and Carstens and Wefer, 1992). Of course, there are many more studies located in the marginal ice zone/seasonal ice zone.

*Line 107: The 200-100 m water depth interval needs to be included.*

Thank you for spotting this, we corrected this.

*Line 108: No 'cod ends' but 'sampling cups'.*

We adopted the change for better reading.

*Line 164: Were the nutrients analyzed from filtered or unfiltered seawater?*

unfiltered

*Various lines: Change maximal to maximum*

We adopted the change

*E.g., lines 237-240: change to past tense. In general, please be careful with past and present tense, etc.*

We corrected the tense

*In general, numbers without units (except of salinity) under 12 should be spelled out.*

Ok, changed

*Figure 5: Please consider making all scales the same length to allow for comparability.*

In this figure, we opted to use different scales in order to optimally visualise the variability down the water column (the purpose of this figure). In Figure 6, the abundances can be compared across the sites.

*Lines 287-290, and line 315: Did you take the time of sampling and synodic lunar reproduction cycle into consideration, which may have affected the size distribution? See Schiebel et al. (2017). Information of the reproduction of N. pachyderma may be obtained from the test-size distribution of assemblage. Small and large tests may indicate a living population, i.e., reproduction and growth.*

This is a good point which we had indeed been thinking about, but it was rather difficult to investigate this, given the limited number of sampling events in our study. We have another study in the pipeline, discussing reproduction of *N. pachyderma* in detail, including some remarkable coiling data. We feel it is better to thoroughly discuss the reproduction there, than as a sidenote in this paper.

*Line 295: How do you assess reproduction? Did you find any gametogenic calcite?*

Yes, these tests had gametogenic calcite, giving *N. pachyderma* it's typical "thick-shelled" character  (i.e. the form mostly present in the sediment).

*Lines 299-301: It would be interesting to analyze the contents of the food vacuoles to determine the different types of algae used as food source.*

Absolutely, we have another study in the pipeline that focuses on their diet (microbiome analysis performed by Clare Bird).

*Line 379: Please see also the papers of Brunner and Biscaye (2003) and Retailleau et al. (2009, 2011, 2012) for neritic effects.*

Interesting, we read the papers and cited these.

*Figure 9: Some of the stations are the same in the left and right panel. Better combine the two panels and use different markers / colours for the different types of stations?*

We adopted the suggested changes

*Figure 11: the legend is too small to be easily read, and should be presented in a different kind of way.*

We now explained the box plots in the caption.

*Table 1: Change 'Depth net' to 'Sampling depth interval (m)'*

We adopted the change

*Plates: Are all scale bars the same length?*

Yes, we now clarified this in the caption

The authors provide quantitative data on the abundance and size of planktonic foraminifera in the upper water layer in the high Arctic, including at stations sampled underneath perennial sea ice. The sampling was carried out by net with fine mesh sizes and followed a depth stratified pattern and the resulting observations are therefore the first ones of its kind. As the authors note, the dataset will represent an important benchmark for the monitoring of the ongoing Atlantification of the high Arctic and the observed distributional patterns will be important for paleoceanographers analysing sedimentary records with polar planktonic foraminifera. Although there are some minor issues with the methodology and interpretation, which I outline below, the study as a whole is well conceived and appropriately documented and I recommend publication with minor revisions.

We thank prof. Michal Kucera for his useful review, in which he points out some issues which we realise we had not been clear about and/or were previously unsure how to address. We have followed these recommendations and changed the manuscript accordingly.

**Comparing foraminifera from samples taken with different mesh size**

Planktonic foraminifera increase the size of their shell by a factor of almost 100 throughout life. Similarly, small species and large species reach adult sizes that differ by more than a factor of 10. Considering this, there will always be two factors that hamper comparisons of absolute concentrations across studies and regions. First, the results will be affected by the mesh size that is used for sampling. This was noticed already by Berger (1969, 1971), who introduced quantitative correction factors to convert abundances among different mesh sizes. For comparison of foraminifera concentrations with other studies (e.g., Line 125), considering the different mesh sizes that they used, would it not be worth exploring what the effect of the conversion factors from Berger would be? This would allow a more direct comparison with the only other study from permanently ice covered region by Be (1960).

We absolutely agree with the tremendous importance of mesh size and implications for comparability (which is also why we pointed this out in our introduction). It is indeed interesting to consider the Berger conversion factor, and we have now introduced it into the manuscript – see lines 524-579.

Second, if the present population of planktonic foraminifera in any sample contains man specimens that belong to the same reproductive cohort (being the result of the same reproductive event), then the observed size distribution also reflects the temporal distance from the reproductive event. I.e., shortly after the reproductive event, the foraminifera would be smaller. Why not using the observations from the multiple sites to see if the size distribution changes through time? This will help us to constrain if there was a cohort-like reproduction affecting the entire region or if it was either very local or not present at all, like in the study by Meilland et al. (2021).

This is an interesting point. We have now added a figure showing the changes of the mean maximum diameter throughout the expedition (for the depth intervals in the top 200m), see the new Fig. 12 and lines 631-635. There appear to be no significant trends in the maximum diameters, and so there appears to be no evidence of a synchronised regional reproduction event determining the trend of test size throughout the expedition.

**Distinguishing between empty and cytoplasm-filled shells**

From the methodology section of the paper (Line 112) it is not clear if empty and filled shells were consistently counted separately? This makes it difficult to interpret the observed patterns of abundance and size change with depth. It appears that the authors only counted empty and filled shells separately at one station (Fig. 8). If this is the case then one would expect at least a discussion of the effect of this simplification on the results. For exampling, knowing where large empty shells occur in the water column would provide the necessary constrain on speculations that the authors make about ontogenetic vertical migration in section 4.3.

> We realise that we had not been fully clear whether a systematic distinction was made between cytoplasm-bearing and empty specimens (see also response to reviewer #1). Indeed, a systematic counting of cytoplasm content was only performed at one station (SO21-26-10). We now pint this out in lined 138-143. Our observations while picking the specimens at other stations, although not quantitative, seemed to confirm the general pattern found in SO21-26-10, namely that the bigger /thicker shells at greater depths were indeed generally empty. Vice versa, we observed that the thinner, smaller shells in the shallower samples were virtually all cytoplasm bearing. We have adjusted the methods section to be clearer about this, as well as pointing out in the discussion that our (tentative) interpretations are only based on one station. We also added a new panel to Fig. 9, showing the cytoplasm percentages.

**Analysing the controlling factors on population size**

I am afraid that a much more formal analysis then what the authors provide (Line 356) is needed to make use of the rich environmental data that the authors collected. First, if the authors assume, and this seems to be supported by Fig. 8, that the foraminifera inhabit the top 100 m of the water column, then they should use population numbers from at least two depth intervals per site, considering that they also have all environmental data for those depths available. Then, they should carry out an analysis with appropriate statistical tools (for example a generalised linear model) on data transformed in a way that they do not violate the assumptions of the method (for example log-transformation of all abundances). In this way, they could test the effect of all variables they collected, also include variable like depth and distance from ice margin, and formally show which variables have no effect. This will be extremely helpful, even for all the negative results. I am afraid that as it stands, the analysis in Fig. 9 is inappropriate and meaningless. There are too few cases and a selection as made by the authors is dangerous and misleading – one can always find a way by which another group of stations can be distinguished from the other stations and then there may even be a negative trend between abundance and chlorophyll. All of this, by the way, requires a consideration of the reproductive dynamics as described above. We must exclude the possibility that all that we see is a cohort that is getting bigger with time (and thus along the cruise) and thus more abundant in the analysed mesh size interval.

> We agree that experimenting with GLMs can be a useful additional analysis and have now done so, see lines 220-228 and 505-511 and the new Table 3. We have been careful relying on the GLM for extensive interpretations since the number of observations is rather low, and there is a high risk of overfitting (a common rule of thumb is to have at least 10 observations per predictor, meaning that at least 70 observations would be needed if we include all possible predictors shown in Fig. 5 - compared to the 24 observations available from multinets). Nevertheless, we believe this was useful and trust that the added analysis gives a better insight into the data.

**Food source for foraminifera living below permanent sea ice**

This is a very important question and the most obvious unexplained pattern is the large concentration below 50 m, i.e. below the productive zone as indicated by chlorophyll profiles. Either all of the foraminifera in the sampling interval 100-50 m resided just below 50 m and fed on phytoplankton corresponding to the bottom

end of the chlorophyll distribution interval, or they fed on something else. In this context, it would seem relevant for the hypotheses that the authors make to consider the study by Greco et al. (2021), which indicates that *N. pachyderma* likely feeds on aggregates rather than on fresh diatoms.

>We fully agree with this comment and had actually been thinking along the same lines. We will now explicitly add a sentence about the possibility of aggregate-feeding in the 50-100m interval in lines 602We follow the latest international standards (TEOS10) and quote values of absolute salinity, in g/kg1-603.

**Minor comments**

Line 53: here the reference to Darling and Wade (2008) is not appropriate because unlike the other studies quoted, the review paper presents no original observations. I recommend to quote primary literature by the author at this place.

>We've revised the reference to the original observations presented in Darling at al., 2004.

Line 64-66: the sentences should be reformulated, the first one has incorrect grammatical structure and the second one likely confused "were" for "where".

>We reformulated both sentences

Line 299: the data discussed here are not given in Fig 7 (should likely be Fig 8). The wording "dominate" is subjective and for many readers would not be consistent with the data. I recommend to use the more neutral "more abundant", here and elsewhere throughout the paper.

>Agreed, made the adjustments

Line 403: rather than gametogenesis it is safer to refer to "reproduction".

>True,  made the change

Line 380: There is abundant anecdotal and less abundant but solid quantitative evidence that planktonic foraminifera do not inhabit waters shallower than about 100 m. They avoid (or are unable to live in) these waters irrespective of temperature (Puerto Rico shelf, North Sea) or food availability, and this pattern also applies to silled fjords, despite the greater water depth further inland beyond the sill. Thus,

the conclusion at this place that the observed low abundances in the RYDER19 region are due to lack of food is unlikely. Darling et al. (2007) observed a completely foraminifera-free Bering Sea, despite high productivity in that region.

Reviewer 1 made a similar comment, see also the response there. We adjusted the manuscript accordingly, adding that we cannot discern between the influence of primary productivity and bathymetry, and added that foraminifera generally don't inhabit such shallow water depths see the new lines 526-530. We note that planktonic foraminifera do reside in the Bering Sea, but are absent in the very shallow Bering Strait (<50m).

---

## Author Response (AR2)

The paper reads good, but does still include many little flaws, which need to be corrected before the paper can be accepted for publication.

-Salinity still has no unit.

**We are aware that according to the 'old' system, salinity is expressed without unit. This is indeed still the case for *practical salinity*. However, here we report *absolute salinity*, as per the latest recommendations (TEOS10), which is reported in in g/kg.  Therefore, we follow the latest international standards (TEOS10) and quote values of absolute salinity, in g/kg.**

-Planktonic foraminifera may be abbreviated PF in the very beginning of the paper and be used throughout instead of the full term, 'foraminifera' only, or 'planktonic' only.

**We will follow the editorial recommendation on the use of abbreviations. We have kept the unabbreviated form for now since it tends to enhance readability for the broader audience.**

-New papers have been published, which show the invasion of new species in polar waters from the North Atlantic Current entering the Arctic Ocean

**We have read Chaabane et al. (2024), which presents interesting findings that our study complements well. Chaabane et al. showed that some species are moving towards the pole, in our study we show that these have not reached the pole yet. We clearly emphasized that our results relate to the perennially ice-covered part of the Arctic Ocean.**

Lines 65-66, What is suggested by the statement that 'In general, the extent to which T. quinqueloba occurs in the central Arctic Ocean is a topical question.'? What is the topical question?

**We reformulated the sentence**

Figure 1, LS is not shown in the map, AB is not explained in the caption. What are the small green dots with the black line, not shown in legend?

**Fixed**

Line 152, change 'foraminifea' to PF

**Fixed**

Line 190, better put: '…against samples collected from the Niskin bottles of the rosette water sampler.'

**Fixed**

Line 201, do you mean the stern of the ship, not bow?

**No, the CTD is commonly deployed from the bow on Oden.**

Line 219, Anacystis nidulans in italic font

**Fixed**

Lines 234-250, please state which water body is under the AW, because this is included with some of your CTD profiles.

**The deep water, fixed.**

Line 286, 'very thick' is what? Better give number.

**Fixed**

Line 294, NO23 should read NO3 plus NO2

**Multimeter thick, fixed**

Line 296, better 'pronounced chlorophyll-a peak'

**Fixed**

Line 297, 'much weaker' than where or what? Relative clause!

**'Than the central Arctic sites', fixed**

Line 298-299, better 'subsurface chlorophyll maximum'

**Fixed**

Lines 301-306, such a detailed description of the oxygen profiles is not possibly needed, and a word on the general trend plus a reference to the Fig. 5 would be sufficient.

**We decided to keep the description.**

Lines 307-309, 'The nutrient...', this section should be moved to the Discussion and provided with references.

**We believe that, due to its descriptive nature; the sentence belongs in the Results section.**

Line 321, 'Spatial variability' should read 'Regional variabilty', of the section 3.2.2 should be made section 3.2.2.1, because 'Depth variability' is also spatial variability (not temporal)

**Fixed**

Line 329, 'different' from what?

**Clarified**

Figure 5: Please make all x-axes the same length to allow for comparability. As requested earlier.

**We equalized x-axes formatting.**

Lines 381-382, how do you know that empty tests were produced by reproduction? Any sign of GAM calcification? As requested earlier.

**Yes, most of these have gametogenic calcite, but we didn't check this systematically so we've removed the statement "following reproduction":**

Figure 7, I would suggest to skip the figure, which coveys a very simple message in a complicated kind of way, and caused more confusion than support a better understanding of the distribution of T vs. S vs. water depth.

**OK, we've removed the figure**

Discussion section, for N. pachyderma and T. quinqueloba, please also have a look at the paper of Simstich et al. (2003, ecology and isotopes), as suggested earlier, and which may add value to the manuscript.

**Because the suggested paper is a bit more focused towards isotopes we unfortunately didn't find an elegant way to fit into our discussion paragraph.**

Line 440, change 'currents' to 'water bodies'

**Fixed**

Line 468, change 'were able to' to 'may'

**Fixed**

Lines 495-496, please discuss the findings of Chaabane et al. 2024 published in Nature on November 13, 2024.

**We have now disussed Chaabane et al. 2024 in the discussion and conclusions.**

Lines 505-511, the correct reference is always Bé 1960

**Fixed**

Figure 10, should possibly be moved to the Results section

**Yes, we will evaluate this during proofing.**

Figure 11, should possibly be moved to the Results section, and the data should be discussed for the synodic lunar reproduction cycle and size structure of assemblages presented in Schiebel et al. 2017, as suggested earlier, and which may add value to the paper.

**We agree with this – we have a follow study in progress where are discussing synodic lunar reproduction cycle, size structure as well as surprising coiling directions observed in the dataset.**

In reviewing the manuscript by Vermassen et al., The distribution and abundance of planktonic foraminifera under sea-ice in the Arctic Ocean, I read the original submission, the three original reviews, the authors' response to the reviewers, and the revised manuscript. Like the authors and Reviewers 1-3, I find the presented dataset to be a valuable baseline for Atlantification of the Arctic that should be published. I also find that the authors were careful and thorough in their consideration of the reviewers' comments.
Several minor issues remain, however, that require attention before this manuscript should be accepted. Mostly, the manuscript should be revised for clarity and consistency (see my notes below) to include verb tense, use of italics, and spaces before units. As another example, the figures are out of order, and Figure 9 does not exist. Crucially, much of the data is missing from this version. For example, I see under Data Availability that foram species count tables are under revision at the Bolin Centre database, but I don't have access to them. It is not possible to reproduce these results without access to the data, and therefore all the data must be available before publication.
The necessary corrections listed below are relatively minor individually, but as a whole they are critically important to successfully communicate the science presented here.

**We thank Marci Robinson for her thorough review of the manuscript. We have revised the manuscript according to her proposed series of minor/technical corrections. Figure referencing has now been fixed, and the datasets were published online at the Bolin Centre database and are accessible.**

First line in Section 2.2. "One photo was taken for each square on the microfossil slide" makes it sound like all 15,381 individuals were on one slide. Perhaps change "slide" to "slides" and revise this sentence from the previous section, "Planktonic foraminifera individuals were picked onto microfossil slides using a combination of pipettes and brushes", to indicate that picking resulted in one slide per what… per depth range per site? Presumably, each slide represents one sample with identified species counts.

**We clarified that , per depth interval per site, tests were divided over several slides (when needed) and that one image per square was obtained (with the tests being divided over multiple squares).**

Section 2.3. All these data must be available through links provided under Data Availability or as tables in this manuscript. It's not clear to me, though I may be missing it, that the chlorophyll-a and nutrient data are included in the datasets accessed by the links provided.

**As now stated in Data Availability, Foraminiferal data is available at the Bolin Centre Database, Chlorophyll data is available as a supplementary table.**

Section 3.1.1. Improper use of the word "lied". Change this sentence to something like: Below the Atlantic Water, the deep water was characterized by temperatures lower than 0°C but with a salinity that remained high.

**Fixed**.

Section 3.1.2. Replace 'real' with true (without quotes).

**Fixed**

Section 3.3.1. Please check the figure citations in this section. I think Fig. 4 should be 5, and Fig. 3 should be 4.

**The figure citations and referencing were checked and fixed.**

Section 3.2.3. You jump to citing Fig. 11 here after Fig. 6, skipping 7-10. I would delete the sentence about a follow-up study. You don't need quotes around 'empty'.

**The figure citations and referencing were checked and fixed. The sentence was deleted, as well as the quotes.**

Section 4.2. First line reads "In order determine their". Change to "in order to determine the". Also, be consistent throughout the manuscript italicizing the cruise names.

**Fixed**

Section 4.3, line 574. Change "fossil test" to "fossil tests".

**Fixed**

Conclusions, line 640. Plateau should be capitalized in Yermak Plateau.

**Fixed**

References. I did not check the references, but I would recommend a thorough proofreading of references and citations before resubmission.

**References checked**

Figures, tables, and plates
Figures are cited out of order, though it's possible I missed something. I believe Figure 2 is cited before Figure 1; Figure 11 is cited before Figures 7-10, and Figure 7 is not cited at all. Figure 9 does

not exist. Are all the Appendix figures cited in the text? Please check the order of all figures and correct if necessary.

**The figure citations and  referencing of appendices were checked and fixed.**

Figure 1. Should the Ryder19 dots be the same size as the SAS Oden dots? Should they be labled like the SAS Oden dots? The figure caption explains the red arrow on the inset map, but what do the orange arrows represent?

**We homogenized the labels and clarified the arrows**

Figure 2 caption. Use lowercase b) to match the figure and the a). Also, the site numbers in Figure 2 are different from the site numbers in figure 1. What do they mean? How is SO21-26-10 in Figure 1 different than SO21-26 (19/8) in Figure 2?

**Fixed and clarified the site numbers (which are abbreviated here).**

Figure 3. Be consistent with capitalization of As and Bs in the panels and the captions. What are the journal formatting rules for this? Spell out CTD in the caption.

**Fixed and we kept 'CTD' as it is defined previously.**

Figure 5. The data presented here must also be available in data tables, either attached to this article or accessible by links.

**All data is now accessible through links or tables.**

Figure 6. Instead of using "Left" and "Right", use A and B consistent with the other figures. Instead of SAS21, do you mean SAS ODEN21? Change RYDER 2019 to RYDER19 to be consistent with the rest of the manuscript.

**Fixed**

Figure 8. These color data need to be included in a data table.

**Data now available in data Table A1.**

Figure 10. Make sure these data tables are accessible.

**Data now available at Bolin Center Database**

Figure 11. These size data need to be included in a data table.

**Data now available at Bolin Center Database**

Figure 12. Do these data include all samples from all sites from both cruises?

**We clarified that the data relate to SAS ODEN21.**

Check to make sure the tables are cited in order. They should also all be formatted the same way.

**Fixed**

Table 1. Change the blue text to white.

**Fixed**

Table 3. Spell out const, avg and std.

**Fixed**

Plates. The figure name formatting is inconsistent. What are the journal rules for plate figure numbering?

**We homogenized the numbering of the plate elements.**

Supplementary Table S1 requires a caption.

**Fixed**